# Learning vector fields of differential equations on manifolds with geometrically constrained operator-valued kernels

**Daning Huang**
Department of Aerospace Engineering
The Pennsylvania State University
University Park, PA 16802, USA
daning@psu.edu

**Hanyang He**
Department of Electrical Engineering
The Pennsylvania State University
University Park, PA 16802, USA
hfh5310@psu.edu

**John Harlim**
Department of Mathematics,
Department of Meteorology and Atmospheric Science
Institute for Computational and Data Sciences
The Pennsylvania State University
University Park, PA 16802, USA
jharlim@psu.edu

**Yan Li**
Department of Electrical Engineering
The Pennsylvania State University
University Park, PA 16802, USA
yql5925@psu.edu

## Abstract

We address the problem of learning ordinary differential equations (ODEs) on manifolds. Existing machine learning methods, particularly those using neural networks, often struggle with high computational demands. To overcome this issue, we introduce a geometrically constrained operator-valued kernel that allows us to represent vector fields on tangent bundles of smooth manifolds. The construction of the kernel imposes the geometric constraints that are estimated from the data and ensures the computational feasibility for learning high dimensional systems of ODEs. Once the vector fields are estimated, e.g., by the kernel ridge regression, we need an ODE solver that guarantees the solution to stay on (or close to) the manifold. To overcome this issue, we propose a geometry-preserving ODE solver that approximates the exponential maps corresponding to the ODE solutions. We deduce a theoretical error bound for the proposed solver that guarantees the approximate solutions to lie on the manifold in the limit of large data. We verify the effectiveness of the proposed approach on high-dimensional dynamical systems, including the cavity flow problem, the beating and travelling waves in Kuramoto-Sivashinsky equations, and the reaction-diffusion dynamics.

## 1 Introduction

In this paper, we consider the problem of learning ODE whose solutions lie on a manifold, which arises from a wide range of applications, from mechanical multibody systems to electrical circuit simulation and power systems (see the references in Ascher & Petzold (1998); Kunkel (2006)), where the system of ODEs on manifolds is formulated based on Differential-Algebraic Equations (Rheinboldt, 1984). One of the main challenges in this problem is that the underlying manifold (geometric) constraints are not explicitly known and need to be uncovered from the data.

A popular solution to this problem is to employ a nonlinear dimensionality reduction approach, such as autoencoders, to represent the geometrical constraint of the dynamics, i.e., the manifold. A typical strategy is to learn a low-dimensional latent space to represent the original data and learn the dynamics in the latent space; the dynamics are represented using, e.g., a neural-network (NN) based discrete-time mapping (Linot & Graham, 2020), recurrent neural network (Maulik et al., 2021; Vlachas et al., 2022), sequence-to-sequence mapping (Wu et al., 2024), Latent Dynamics

Network (Regazzoni et al., 2024), and SINDy (Fukami et al., 2021; Lin et al., 2024). The main pitfall of this class of methods is that the latent space only provides a global parametrization of the manifold, whose dimension is typically larger than the intrinsic dimension of the manifold, and the dynamics are not guaranteed to learn the exponential maps (for discrete-time model) or vector fields (for continuous-time model) of the manifold. As a result, the predicted trajectories may deviate from the manifold, limiting the long-term prediction accuracy, as we shall see in several numerical examples with NN baselines. Furthermore, the computational cost to train such NN-based nonlinear estimators is known to be expensive. In our numerical comparison with the proposed linear method, we found that while their testing times are comparable, the training time for NN models is over 800 times slower on a simple example with two ambient dimensions.

**Contribution.** This motivates us to develop a linear estimator of the vector fields on the tangent bundles of smooth manifolds. Motivated by SINDy algorithm (Brunton et al., 2016), we construct a geometrically constrained dictionary to represent the unknown vector fields where these constraints will be approximated from the available point cloud data induced by the observed time series of the dynamical systems. Since such a dictionary is subjected to the curse of dimensionality, we employ the standard "kernel trick" to mitigate this issue. Unlike previous works in non-manifold setting with scalar-valued kernels (Baddoo et al., 2022; Yang et al., 2024) and kriging/Gaussian process (Glaz et al., 2010), the geometrical constraints give rise to an operator-valued kernel that leverages the intrinsic dimension of the manifold to enable practical implementation. To numerically integrate the system of ODEs with the estimated vector fields, we devise an ODE solver that guarantees the solutions to be on the manifolds in the limit of large data. We will demonstrate the effectiveness of this approach numerically on several high-dimensional test problems, including the cavity flow problem, the beating and travelling waves in Kuramoto-Sivashinsky equations, and reaction-diffusion dynamics.

**Paper organization.** The remainder of this paper is organized as follows: In Section 2, we discuss the geometrically constrained dictionary, extending the SINDy approach and generalizing it to an operator-valued kernel to mitigate high dimensional problems. In Section 3, we introduce a geometry-preserving time integration scheme, provide an illuminating example that motivate this integrator, and discuss the convergence properties. In Section 4, we discuss closely related approaches that will be used as a baseline to quantify the performance of the proposed approaches documented in Section 5. In Section 6, we give a brief summary. We supplement the paper with five appendices that report the detailed technical numerical tools needed in the algorithm, computational complexity, the theoretical proofs, and additional numerical results.

## 2 GEOMETRICALLY CONSTRAINED DICTIONARY

Consider dynamical systems governed by a system of ODEs,

$$\dot{\mathbf{x}} = \mathbf{f}(\mathbf{x}), \quad \mathbf{x} \in \mathbb{R}^n, \tag{1}$$

where the vector field $\mathbf{f} : \mathbb{R}^n \to \mathbb{R}^n$. The Sparse Identification of Nonlinear Dynamics (SINDy) approach (Wang et al., 2011; Brunton et al., 2016) is to approximate components of $\mathbf{f} = (f^1, \dots, f^n)$ by a sparse regression on a set of appropriate basis functions. Typical choices of basis functions can be polynomials and/or trigonometric functions. An example of polynomial dictionary is,

$$\theta(\mathbf{x}) = \begin{pmatrix} 1 & \mathbf{x}^\top & (\mathbf{x}^2)^\top & \dots \end{pmatrix} \in \mathbb{R}^m$$

where $\mathbf{x}^j := \left\{ x_1^{j_1} x_2^{j_2} \cdots x_n^{j_n} : \forall j = j_1 + j_2 + \dots + j_n \right\}$. Given a set of labeled training data $\{\mathbf{x}_i, \dot{\mathbf{x}}_i\}_{i=1,\dots,N}$, where the subscript index denotes temporal information, $\mathbf{x}_i := \mathbf{x}(t_i)$ and $\dot{\mathbf{x}}_i := \dot{\mathbf{x}}(t_i)$, the SINDy approach is to approximate $f^k(\mathbf{x}) \approx f_\epsilon^k(\mathbf{x}; \hat{\xi}_k) := \theta(\mathbf{x})\hat{\xi}_k$ with coefficients $\hat{\xi}_k \in \mathbb{R}^m$ obtained from solving the following sparse regression problem,

$$\hat{\Xi} = \arg\min_{\Xi} \sum_{i=1}^N \|\dot{\mathbf{x}}_i - \mathbf{f}_\epsilon(\mathbf{x}_i; \Xi)\|^2 + \lambda \|\Xi\|_1. \tag{2}$$

where $\lambda > 0$ is a sparsity parameter, $\mathbf{f}_\epsilon = (f_\epsilon^1, \dots, f_\epsilon^n)$, and $\hat{\Xi} = ((\xi^1)^\top, \dots, (\xi^n)^\top)^\top \in \mathbb{R}^{nm}$; by default $\|\cdot\|$ means $\ell_2$ norm. There are two key issues with this approach as it stands. First, the method is

sensitive to the choice of dictionary. If the space spanned by the dictionary does not encompass the underlying function, the estimated vector field will not be accurate when evaluated on new sample data from the same distribution. Second, this method is computationally impractical as $n$ increases since the size of the dictionary increases (exponentially) as a function of $n$. Particularly, if the dictionary consists of monomials of degree up to $p$, then the size of the dictionary is $m \propto p^{n-1}$.

Let us now focus on the first issue for a class of dynamics where the solutions lie on a $d$-dimensional Riemannian sub-manifold $M \subset \mathbb{R}^n$. In this context, the vector field $\mathbf{f} \in \mathfrak{X}(M)$ is a map $\mathbf{f} : M \to TM$ that identifies the state $\mathbf{x} \in M$ to a vector in the tangent space $\mathbf{f}(\mathbf{x}) \in T_\mathbf{x}M$. Denote the bases of the tangent space $T_{\mathbf{x}_i}M \cong \mathbb{R}^d$ and the normal space as columns of $\mathbf{T}_i \in \mathbb{R}^{n \times d}$ and $\mathbf{N}_i \in \mathbb{R}^{n \times (n-d)}$, respectively. We note that these basis vectors can be identified from point cloud data using the local SVD technique (Donoho & Grimes, 2003; Zhang & Zha, 2004) or higher-order methods (Jiang et al., 2024). See Appendix A for the details. For the remainder of this paper, we denote $\hat{\mathbf{T}}_i$ and $\hat{\mathbf{N}}_i$ to be the point cloud approximation to $\mathbf{T}_i$ and $\mathbf{N}_i$, respectively.

Let the matrix $P(\mathbf{x}) : \mathbb{R}^n \to T_\mathbf{x}M \subset \mathbb{R}^n$ be an orthogonal projection to the local tangent space at $\mathbf{x}$. One can show that $P(\mathbf{x}_i) = \mathbf{T}_i \mathbf{T}_i^\top$, where columns of $\mathbf{T}_i \in \mathbb{R}^{n \times d}$ forms a set of orthonormal vectors that span $T_{\mathbf{x}_i}M$. With this background, since $\mathbf{f}(\mathbf{x}) \in T_\mathbf{x}M$, it is clear that $P(\mathbf{x})\mathbf{f}(\mathbf{x}) = \mathbf{f}(\mathbf{x})$ under the $n-$dimensional Euclidean inner product. Practically, when the manifold is unknown, one can approximate $P(\mathbf{x}_i)$ by $\hat{P}(\mathbf{x}_i) = \hat{\mathbf{T}}_i \hat{\mathbf{T}}_i^\top$. Based on this information, we propose the following modification on the SINDy dictionary for modeling the vector field,

$$\mathbf{f}(\mathbf{x}) \approx \mathbf{f}_c(\mathbf{x}; \hat{\Xi}) = \hat{P}(\mathbf{x})\Theta(\mathbf{x})\hat{\Xi}, \tag{3}$$

where $\Theta(\mathbf{x}) \in \mathbb{R}^{n \times nm}$ is a block diagonal matrix with $\theta(\mathbf{x})$ as the diagonal block component, and the coefficients $\hat{\Xi} \in \mathbb{R}^{nm}$ are obtained by fitting the model to the observed vector field, $\dot{\mathbf{x}}_i$. In practice, when the available data are only the time series $X = \{\mathbf{x}_i\}_{i=1}^N$, one needs to approximate the derivatives. We consider approximating $\dot{\mathbf{x}}_i$ with $\mathbf{y}_i = \hat{\mathbf{T}}_i \hat{\mathbf{T}}_i^\top (\mathbf{x}_{i+1} - \mathbf{x}_i)/\Delta t$.

As we mentioned before, another challenge with this dictionary is that it is subjected to the curse of dimensionality, that the number of candidate functions grows exponentially and eventually becomes computationally intractable as the dimension of states increases. To mitigate this problem, we propose an operator-valued kernel deduced from the dictionary in (3) that allows the vector field to lie on the tangent bundle in the limit of large data with a compact model having a rank equal to the intrinsic dimension of the manifold.

In the remainder of this section, we first use the kernel trick to motivate a Geometrically constrained Multivariate Kernel Ridge Regression (GMKRR) model in the ambient space. Then we formalize the GMKRR model rigorously using the Reproducing Kernel Hilbert Space (RKHS) with a family of operator-valued kernels. Lastly, we manipulate the GMKRR model into the intrinsic space to enable practical computational implementation.

## 2.1 Kernelization of the geometrically constrained dictionary

While kernel regression with $\ell_q$ regularization ($0 < q \le 1$) has been studied extensively (see Shi et al., 2019, and the references therein), it is computationally much simpler to employ $\ell_2$ regularization, which will be the focus in this paper. Specifically, we focus on the following problem modifying (2) as the primal form,

$$\hat{\Xi} = \arg\min_{\Xi} \left\| \mathbf{y} - \Psi\Xi \right\|^2 + \lambda \|\Xi\|^2, \tag{4}$$

where $\mathbf{y} = [\mathbf{y}_1^\top, \mathbf{y}_2^\top, \cdots, \mathbf{y}_N^\top]^\top \in \mathbb{R}^{nN}$ with $\mathbf{y}_i = \hat{\mathbf{T}}_i \hat{\mathbf{T}}_i^\top (\mathbf{x}_{i+1} - \mathbf{x}_i)/\Delta t$, and $\Psi = [\psi_1^\top, \psi_2^\top, \cdots, \psi_N^\top]^\top \in \mathbb{R}^{nN \times nm}$ with $\psi_i = \psi(\mathbf{x}_i) = \hat{P}(\mathbf{x}_i)\Theta(\mathbf{x}_i) \in \mathbb{R}^{n \times nm}$.

Next, introduce the dual variable $\alpha \in \mathbb{R}^{nN}$ so that $\Xi = \Psi^\top \alpha$, and the dual form of (4) is

$$\hat{\alpha} = \arg\min_{\alpha} \left\| \mathbf{y} - \Psi\Psi^\top \alpha \right\|^2 + \lambda \left\| \Psi^\top \alpha \right\|^2 \equiv \arg\min_{\alpha} \left\| \mathbf{y} - \mathbf{K}\alpha \right\|^2 + \lambda \alpha^\top \mathbf{K}\alpha \tag{5}$$

where the gram matrix $\mathbf{K} = \Psi\Psi^\top \in \mathbb{R}^{nN \times nN}$ and its $(i, j)$th block is $\mathbf{K}_{ij} = \psi(\mathbf{x}_i)\psi(\mathbf{x}_j)^\top \equiv k(\mathbf{x}_i, \mathbf{x}_j) \in \mathbb{R}^{n \times n}$. The solution to the dual form is $\alpha^* = (\lambda\mathbf{I} + \mathbf{K})^{-1}\mathbf{y}$, and the predictive model for a new input $\mathbf{x}$ is given by,

$$\mathbf{f}_c(\mathbf{x}) = \psi(\mathbf{x})\Xi = \psi(\mathbf{x})\Psi^\top \alpha = \mathbf{k}(\mathbf{x})(\lambda\mathbf{I} + \mathbf{K})^{-1}\mathbf{y} \tag{6}$$

where $\mathbf{k}(\mathbf{x}) = [k(\mathbf{x}, \mathbf{x}_1), k(\mathbf{x}, \mathbf{x}_2), \cdots, k(\mathbf{x}, \mathbf{x}_N)]$. Since $\boldsymbol{\psi}(\mathbf{x}) = \hat{P}(\mathbf{x})\Theta(\mathbf{x}) \in \mathbb{R}^{n \times nm}$, we have

$$k(\mathbf{x}, \mathbf{x}') = \boldsymbol{\psi}(\mathbf{x})\boldsymbol{\psi}(\mathbf{x}')^{\top} = \hat{P}(\mathbf{x})\Theta(\mathbf{x})\Theta(\mathbf{x}')^{\top}\hat{P}(\mathbf{x}') = \rho(\mathbf{x}, \mathbf{x}')\hat{P}(\mathbf{x})\hat{P}(\mathbf{x}') \tag{7}$$

where $\rho(\mathbf{x}, \mathbf{x}') = \theta(\mathbf{x})\theta(\mathbf{x}')^{\top} \in \mathbb{R}$, and the last equality is because $\Theta(\mathbf{x})\Theta(\mathbf{x}')^{\top} = \mathrm{diag}[\rho(\mathbf{x}, \mathbf{x}'), \cdots, \rho(\mathbf{x}, \mathbf{x}')] = \rho(\mathbf{x}, \mathbf{x}')\mathbf{I} \in \mathbb{R}^{n \times n}$.

Up to this point the regression problem with (3) is converted to a GMKRR problem. The geometrically-constrained function $k$ in (7) is used as a matrix-valued kernel and constructed from a finite set of candidate functions defined in the ambient space.

## 2.2 THE RKHS OF THE INTRINSIC GMKRR MODEL

In the following, we generalize the GMKRR model to a family of matrix kernel functions that may include infinitely many candidate functions via the construction of a $\mathfrak{X}(M)$-valued RKHS $\mathcal{H}$.

**Definition 2.1.** *Let $X$ be a non-empty set, $\mathcal{W}$ a separable Hilbert space with inner product $\langle \cdot, \cdot \rangle$, and $\mathcal{L}(\mathcal{W})$ a Banach space of bounded linear operators on $\mathcal{W}$. A function $k : X \times X \mapsto \mathcal{L}(\mathcal{W})$ is SPD if (1) for any pair $(\mathbf{x}, \mathbf{x}') \in X \times X$, $k(\mathbf{x}, \mathbf{x}')^* = k(\mathbf{x}', \mathbf{x})$, and (2) for any finite set of points $\{\mathbf{x}_i\}_{i=1}^{N}$ in $X$ and $\{\mathbf{f}_i\}_{i=1}^{N}$ in $\mathcal{W}$, $\sum_{i,j=1}^{N} \langle \mathbf{f}_i, k(\mathbf{x}_i, \mathbf{x}_j)\mathbf{f}_j \rangle \geq 0$. The function $k$ is an operator-valued kernel on $X$ and $\mathcal{W}$.*

**Definition 2.2.** *Following the notation from the previous definition, for each $\mathbf{x} \in X$ and $\mathbf{f}, \mathbf{g} \in \mathcal{W}$, define $k_{\mathbf{x}}\mathbf{f}(\mathbf{x}') = k(\mathbf{x}, \mathbf{x}')\mathbf{f}$ for all $\mathbf{x}' \in X$. For $\mathbf{f}' = \sum_{i=1}^{N} k_{\mathbf{x}_i}\mathbf{f}_i$ and $\mathbf{g}' = \sum_{i=1}^{N} k_{\mathbf{x}'_i}\mathbf{g}_i$, define the inner product, $\langle \mathbf{f}', \mathbf{g}' \rangle_{\mathcal{H}} = \sum_{i,j=1}^{N} \langle \mathbf{f}_i, k(\mathbf{x}_i, \mathbf{x}'_j)\mathbf{g}_j \rangle$. Then $\mathcal{H} = \overline{\mathrm{span}}\{k_{\mathbf{x}}\mathbf{f} | \mathbf{x} \in X, \mathbf{f} \in \mathcal{W}\}$ forms an RKHS with reproducing kernel $k$. The RKHS has the reproducing property that $\langle \mathbf{f}(\mathbf{x}), \mathbf{g} \rangle_{\mathcal{W}} = \langle \mathbf{f}(\cdot), k(\cdot, \mathbf{x})\mathbf{g} \rangle_{\mathcal{H}}$.*

**Definition 2.3.** *The inner product $\langle \cdot, \cdot \rangle_{\mathcal{H}}$ also induces the RKHS norm, $\|\mathbf{f}'\|_{\mathcal{H}} = \sqrt{\langle \mathbf{f}', \mathbf{f}' \rangle_{\mathcal{H}}}$ for all $\mathbf{f}' = \sum_{i=1}^{N} k_{\mathbf{x}_i}\mathbf{f}_i$. When $\mathcal{W}$ is a $n$-dimensional Euclidean space, $\|\mathbf{f}'\|_{\mathcal{H}} = \sqrt{\mathbf{f}^{\top}\mathbf{K}\mathbf{f}}$, where $\mathbf{f} = [\mathbf{f}_1^{\top}, \mathbf{f}_2^{\top}, \cdots, \mathbf{f}_N^{\top}]^{\top}$ and $\mathbf{K} \in \mathbb{R}^{nN \times nN}$ with the $(i, j)$th block $\mathbf{K}_{ij} = k(\mathbf{x}_i, \mathbf{x}_j)$.*

**Lemma 2.1.** *Consider a function $k : M \times M \mapsto \mathcal{L}(\mathfrak{X}(M))$, defined as $k(\mathbf{x}, \mathbf{x}') = \rho(\mathbf{x}, \mathbf{x}')\hat{P}(\mathbf{x})\hat{P}(\mathbf{x}')$, where $\rho : \mathbb{R}^n \times \mathbb{R}^n \mapsto \mathbb{R}$ is a scalar-valued kernel. Then $k$ is an operator-valued kernel.*

See Appendix D for the proof of Lemma 2.1. The operator-valued kernel $k$ forms the desired $\mathfrak{X}(M)$-valued RKHS, denoted as $\mathcal{H}_M$. In practice, we can use any SPD kernels such as the squared exponential (SE) kernel $\rho(\mathbf{x}, \mathbf{x}') = \exp\left(-\|\mathbf{x} - \mathbf{x}'\|^2 / \gamma\right)$ or the Matérn kernels (see Appendix E.1).

The function in (7) is a special case of the operator-valued kernel on $M$ and $\mathfrak{X}(M)$. Subsequently, the GMKRR model is reformulated via $\mathcal{H}_M$. Given a dataset $\{(\mathbf{x}_i, \mathbf{y}_i)\}_{i=1}^{N}$, the unknown vector field $\mathbf{f} \in \mathcal{H}_M$ is parametrized as $\mathbf{f}(\mathbf{x}) = \sum_{i=1}^{N} k(\mathbf{x}_i, \mathbf{x})\boldsymbol{\alpha}_i \equiv \mathbf{k}(\mathbf{x})\boldsymbol{\alpha}$, where $\boldsymbol{\alpha} = [\boldsymbol{\alpha}_1^{\top}, \boldsymbol{\alpha}_2^{\top}, \cdots, \boldsymbol{\alpha}_N^{\top}]^{\top}$ is determined by minimizing the following objective function,

$$J(\mathbf{f}) = \sum_{i=1}^{N} \|\mathbf{y}_i - \mathbf{f}(\mathbf{x}_i)\|^2 + \lambda \|\mathbf{f}\|_{\mathcal{H}_M}^2 \equiv \|\mathbf{y} - \mathbf{K}\boldsymbol{\alpha}\|^2 + \lambda \boldsymbol{\alpha}^{\top}\mathbf{K}\boldsymbol{\alpha},$$

which is the same optimization problem solved in the previous section, and the solution is given by (6). However, now the new formulation admits a family of operator-valued kernels, that may involve an infinite set of candidate functions.

## 2.3 CONVERSION TO INTRINSIC SPACE

Subsequently, we formulate the GMKRR model (6) so that the predictive model is effectively defined in the intrinsic space and becomes computationally tractable to train and evaluate. Note that $\hat{P}(\mathbf{x}) = \hat{\mathbf{T}}_{\mathbf{x}}\hat{\mathbf{T}}_{\mathbf{x}}^{\top}$, the operator-valued kernel $k$ in ambient space can be rewritten as

$$k(\mathbf{x}, \mathbf{x}') = \rho(\mathbf{x}, \mathbf{x}')\hat{P}(\mathbf{x})\hat{P}(\mathbf{x}') = \rho(\mathbf{x}, \mathbf{x}')\hat{\mathbf{T}}_{\mathbf{x}}O_{\mathbf{x}\mathbf{x}'}\hat{\mathbf{T}}_{\mathbf{x}'}^{\top} \equiv \hat{\mathbf{T}}_{\mathbf{x}} r(\mathbf{x}, \mathbf{x}')\hat{\mathbf{T}}_{\mathbf{x}'}^{\top},$$

where $O_{\mathbf{x}\mathbf{x}'} = \hat{\mathbf{T}}_{\mathbf{x}}^{\top}\hat{\mathbf{T}}_{\mathbf{x}'} \in \mathbb{R}^{d \times d}$, and

$$r(\mathbf{x}, \mathbf{x}') = \rho(\mathbf{x}, \mathbf{x}')\hat{\mathbf{T}}_{\mathbf{x}}^{\top}\hat{\mathbf{T}}_{\mathbf{x}'} = \rho(\mathbf{x}, \mathbf{x}')O_{\mathbf{x}\mathbf{x}'} \in \mathbb{R}^{d \times d}. \tag{8}$$

Using (8), the gram matrix $\mathbf{K}$ in ambient space is decomposed as $\mathbf{K} = \mathcal{T}\mathbf{R}\mathcal{T}^\top$, where $\mathcal{T} \in \mathbb{R}^{nN \times dN}$ is a block diagonal matrix with diagonal block entries $\hat{\mathbf{T}}_1, \hat{\mathbf{T}}_2, \ldots, \hat{\mathbf{T}}_N$, and $\mathbf{R} \in \mathbb{R}^{dN \times dN}$ is a $N \times N$ block matrix with the $(i,j)$th block $\mathbf{R}_{ij} = r(\mathbf{x}_i, \mathbf{x}_j) = \rho(\mathbf{x}_i, \mathbf{x}_j)O_{\mathbf{x}_i\mathbf{x}_j}$. Similarly,

$$\mathbf{k}(\mathbf{x}) = \hat{\mathbf{T}}_{\mathbf{x}}[r(\mathbf{x}, \mathbf{x}_1), r(\mathbf{x}, \mathbf{x}_2), \cdots, r(\mathbf{x}, \mathbf{x}_N)]\mathcal{T}^\top \equiv \hat{\mathbf{T}}_{\mathbf{x}}\mathbf{r}(\mathbf{x})\mathcal{T}^\top. \tag{9}$$

Using the above decompositions, the GMKRR formulation (6) in ambient space is converted to,

$$\mathbf{f}_\epsilon(\mathbf{x}) = \mathbf{k}(\mathbf{x})(\lambda\mathbf{I} + \mathbf{K})^{-1}\mathbf{y} = \hat{\mathbf{T}}_{\mathbf{x}}\mathbf{r}(\mathbf{x})\mathcal{T}^\top\left(\lambda\mathbf{I} + \mathcal{T}\mathbf{R}\mathcal{T}^\top\right)^{-1}\mathbf{y} = \hat{\mathbf{T}}_{\mathbf{x}}\mathbf{r}(\mathbf{x})\left(\lambda\mathbf{I} + \mathbf{R}\right)^{-1}\mathcal{T}^\top\mathbf{y} \tag{10}$$

where the Woodbury identity is used in the last equality.

In the intrinsic GMKRR formulation (10), the matrix $(\lambda\mathbf{I} + \mathbf{R})$ is of dimension $dN \times dN$ and computationally tractable to invert, especially if $d$ is small; this is regardless of the ambient dimension $n$. Furthermore, the term $\hat{\mathbf{T}}_{\mathbf{x}}$ guarantees that the vector fields lie on the local tangent space of the underlying manifold at $\mathbf{x}$ in the limit of large data.

Lastly, we briefly discuss the intrinsic GMKRR model from an RKHS point of view. First, it can be proved that the function $r$ in (8) is an operator-valued kernel on $M$ and $\mathcal{L}(\mathbb{R}^d)$ (the proof is similar to that of Lemma 2.1); $r$ is referred to as the intrinsic operator-valued kernel. Then, $r$ induces an RKHS and the corresponding GMKRR model in the intrinsic space. The GMKRR is effectively applied to a modified dataset $\{(\mathbf{x}_i, \bar{\mathbf{y}}_i = \hat{\mathbf{T}}_{\mathbf{x}_i}^\top\mathbf{y}_i)\}_{i=1}^N$, where the modified label $\bar{\mathbf{y}}_i$ is the vector field expressed in the local tangent space at $\mathbf{x}_i$. In the kernel $r(\mathbf{x}, \mathbf{x}')$, if (1) $\rho$ is chosen to be the Diffusion Map kernel and (2) the pairs of data points $(\mathbf{x}, \mathbf{x}')$ are sufficiently close so that $O_{\mathbf{x}\mathbf{x}'} = \hat{\mathbf{T}}_{\mathbf{x}}^\top\hat{\mathbf{T}}_{\mathbf{x}'}$ is always orthogonal, then the RKHS induced by $r$ is a subset of $L^2(\mathfrak{X}(M))$ that is spanned by smooth eigenvector-fields of the Connection Laplacians (Singer & Wu, 2012).

## 3 GEOMETRY-PRESERVING TIME INTEGRATOR

While standard ODE solvers such as Runge-Kutta methods often empirically produce solutions that are close enough to the manifold (i.e., a manifold invariant scheme) for sufficiently small time step, it is well known that the invariant manifold property is only valid when the solvers are employed on a special class of manifolds (Calvo et al., 1996). Various ODE solvers on manifolds have been proposed in literature, see Hairer (2011); Crouch & Grossman (1993) for general vector fields and Leimkuhler & Patrick (1996) for Hamiltonian systems. In this section, we will illustrate this issue on a simple example, propose a normal correction (NC) to the classical explicit Euler scheme which we call Euler+NC in the remainder of this paper, and provide a convergence study. This approach can be viewed as a realization of the local coordinate approach (see Section III.2 in Hairer, 2011) with local parameterization being estimated by GMLS. The proposed normal correction approximates all of the higher-order terms in an exponential map, $\exp_{\mathbf{x}_i}(\mathbf{f}(\mathbf{x}_i)\Delta t)$, including the second fundamental form, and is computationally more attractive than the classical Taylor's method that requires the derivatives of the estimated vector fields.

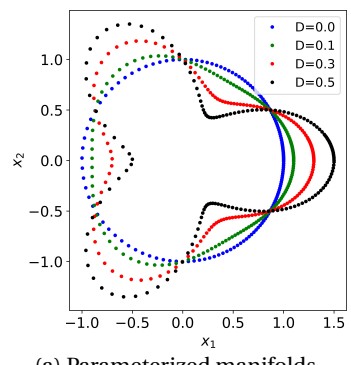

(a) Parameterized manifolds.

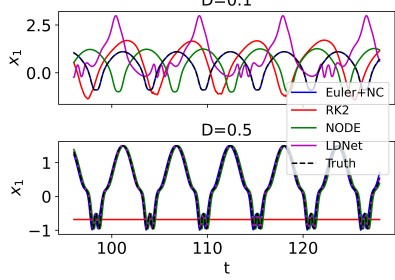

(b) Predictions at a long time period.

Figure 1: Dynamics on a series of 1D manifolds.

### 3.1 A MOTIVATING EXAMPLE

Consider a scalar ODE: $\dot{\theta} = \frac{3}{2} - \cos(\theta)$, $\theta(0) = 0$, whose solution has a period of $2\pi$, and embed the solution in a 2D ambient space by $(x_1, x_2) = (r(\theta)\cos(\theta), r(\theta)\sin(\theta))$, where $r(\theta) = 1 + D\cos(K\theta)$. The 2D embedding is illustrated in Fig. 1a for $K = 3$ and a series of $D$ values, where the neighboring points separate apart by one step size $\Delta t = 0.04$. When $D = 0$, the 1D manifold is a "simple" unit circle. As $D$ increases, the manifold becomes more distorted, which may pose a challenge in solving the dynamics.

For each value of $D$, a GMKRR model is trained using the trajectory data obtained in the ambient space (details in Appendix E.1). Subsequently, the GMKRR model is solved using standard RK2 and the proposed Euler+NC method. Meanwhile, a neural ODE (NODE) model (Chen et al., 2018) and a Latent Dynamics Networks (LDNet) model (Regazzoni et al., 2024) are trained as baselines. In Fig. 1b, we show the prediction quality for $x_1$ at time interval $[96, 128]$ for $D = 0.1$ and $0.5$. From Fig. 1b, we note that the proposed Euler+NC consistently preserves the geometrical constraint; yet, RK2 solutions deviate from the truth. The trajectories of both NODE and LDNet are off the manifold for $D = 0.1$. For $D = 0.5$, the NODE achieves a similar accuracy as GMKRR with Euler+NC, while LDNet is very accurate (see Table 3 for the RMSE comparison). Both NODE and Euler+NC develop phase shifts where the former is less accurate relative to the latter. This result suggests the robustness of the proposed solver over the nonlinear methods, that are much more expensive to train. Specifically, for LDNets, the results above are obtained using twice the size of training data compared to others (we fail to train competitive LDNet using the same data size) and we also found that LDNet with 1D latent variable (which is the intrinsic dimension for this problem) does not work. See the complete results in Appendix E.1, where we also reported simulations for other values of $D$, their root-mean-square-errors, comparisons of training and prediction times, comparisons of results with Gaussian and Matérn kernels for $\rho$, and the numerical verification of the theoretical convergence rate reported in Theorem 3.1.

## 3.2 EULER METHOD WITH NORMAL CORRECTION

To solve ODEs on manifold, we consider an integrator based on the following identity:

$$\mathbf{x}_{i+1} = \exp_{\mathbf{x}_i}(\mathbf{f}(\mathbf{x}_i)\Delta t)$$
$$= \mathbf{x}_i + \underbrace{\mathbf{T}_i\mathbf{T}_i^\top(\mathbf{x}_{i+1} - \mathbf{x}_i)}_{\mathbf{f}(\mathbf{x}_i)\Delta t} + \underbrace{\mathbf{N}_i\mathbf{N}_i^\top(\mathbf{x}_{i+1} - \mathbf{x}_i)}_{O((\Delta t)^2)}, \quad (11)$$

where the exponential map $\exp_{\mathbf{x}_i} : T_{\mathbf{x}_i}(M) \to M$ is defined as a map of the vector field $\mathbf{f}$ along the local tangent space $T_{\mathbf{x}_i}M$ plus a second-order correction in the normal direction, which can be approximated as a function of $\mathbf{f}$ (see Figure 2).

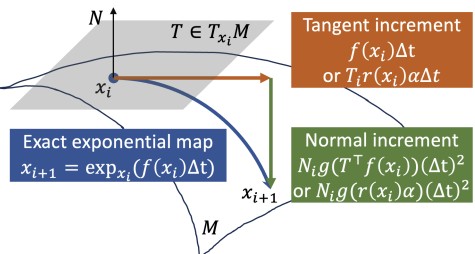

Figure 2: Dynamics decomposed on the manifold.

Based on the identity in (11), once $\mathbf{f}_\epsilon$ is trained (e.g, by the GMKRR model in (10)), the solution $\mathbf{x}_\epsilon(t_{i+1})$ to the following approximate differential system,

$$\dot{\mathbf{x}}_\epsilon(t) = \mathbf{f}_\epsilon(\mathbf{x}_\epsilon(t)), \quad t > t_0, \quad \mathbf{x}_\epsilon(t_0) = \hat{\mathbf{x}}_{\epsilon,i}$$

is approximated by $\hat{\mathbf{x}}_{\epsilon,i+1}$ through the following steps:

1. Use the algorithm in Appendix A to approximate bases of the local tangent and normal spaces of $\hat{\mathbf{x}}_{\epsilon,i}$. We denote the estimates as $\hat{\mathbf{T}}_{\epsilon,i}$ and $\hat{\mathbf{N}}_{\epsilon,i}$.

2. Let $\{\mathbf{x}_{i,1}, \ldots, \mathbf{x}_{i,k}\} \subset X$ be the $k$th nearest neighbors of $\hat{\mathbf{x}}_{\epsilon,i}$. We model the normal component of (11) by solving the following local regression problem,

$$\hat{g} := \arg\min_g \sum_{j=1}^k \|\hat{\mathbf{N}}_{\epsilon,i}^\top(\mathbf{x}_{i,j} - \hat{\mathbf{x}}_{\epsilon,i}) - (\Delta t)^2 g(\hat{\mathbf{T}}_{\epsilon,i}^\top(\mathbf{x}_{i,j} - \hat{\mathbf{x}}_{\epsilon,i})/\Delta t)\|^2. \quad (12)$$

Notice that $\hat{g} : \mathbb{R}^d \to \mathbb{R}^{n-d}$ takes input from the local coordinates basis $\hat{\mathbf{T}}_{\epsilon,i}$. To solve this regression problem, we will employ the Generalized Moving Least Squares (GMLS) method that is documented in Appendix A. In Appendix D.2, one can see that $\hat{g}$ approximates the second fundamental form of $M$ up to order-$\Delta t$.

3. We predict the state at time $t_{i+1} = t_i + \Delta t$ with the following model that approximates (11),

$$\hat{\mathbf{x}}_{\epsilon,i+1} = \hat{\mathbf{x}}_{\epsilon,i} + \Delta t \mathbf{f}_\epsilon(\hat{\mathbf{x}}_{\epsilon,i}) + (\Delta t)^2 \hat{\mathbf{N}}_{\epsilon,i} \hat{g}(\hat{\mathbf{T}}_{\epsilon,i}^\top \mathbf{f}_\epsilon(\hat{\mathbf{x}}_{\epsilon,i}))). \quad (13)$$

Here, we simply evaluate $\hat{g}$ as illustrated in Figure 2.

We should point out that the proposed Euler+NC method has approximately the same complexity as RK2, as RK2 involves two function evaluations and each requires one local least-squares fitting to approximate $\hat{\mathbf{T}}$. In Euler+NC, we also employ two least-squares fittings, one in Step 1 and the other in Step 2. See Appendix C for the complexity of the three steps above.

### 3.3 Convergence theory

In the following result, we state the error bounds predicting the state $\mathbf{x}_n = \mathbf{x}(n\Delta t) = \mathbf{x}(T)$ for some fixed $T > 0$ with a general approximate vector field $\mathbf{f}_\epsilon$ under the numerical integrator (13). In this derivation, we assume that the local tangent and normal spaces are approximated with the GMLS method discussed in Appendix A.3 with a polynomial of degree-$\ell$, provided that the manifold $M$ is sufficiently smooth, $C^{\ell+1}$, with $\ell \geq 2$. Simultaneously, $\hat{g}$, attained by solving (12), is also a polynomial of order-$\ell$.

**Theorem 3.1.** *Let the solution of* (1) *lie on a smooth (or at least $C^{\ell+1}$) manifold $M$, with $\ell \geq 2$. Let $\mathbf{f}_\epsilon$ be a Lipschitz function that approximates $\mathbf{f}$, such that $\|\mathbf{f} - \mathbf{f}_\epsilon\|_{L^2(\mu)} = O(\epsilon)$ as $\epsilon \to 0$. Here, $\mu$ denotes an invariant measure of* (22). *Let $X = \{\mathbf{x}_1, \ldots, \mathbf{x}_N\}$ be independent and identically distributed (i.i.d.) samples of $\mu$ that is absolutely continuous with respect to the volume measure, $d\mu = q\, d\text{Vol}$, with $0 < q_{min} \leq q \leq q_{max}$. Then with probability higher than $1 - \frac{3}{N}$,*

$$\sqrt{\mathbb{E}_\mu\left[\|\mathbf{x}_n - \hat{\mathbf{x}}_{\epsilon,n}\|^2\right]} \leq C\left(\epsilon + \Delta t\left(\frac{\log(N)}{N}\right)^{\frac{\ell}{d}}\right),$$

*where $T = n\Delta t$. Here the constant $C > 0$ is independent of $N$ and $\Delta t$. See Proposition D.2 for the detailed expression of this constant.*

See Appendix D.2 for the proof. We should point out that $\epsilon$ is the bias, which can be specified further in terms of $N, \ell, d$, depending on the class of models being used to approximate $\mathbf{f}$. For example, error bounds for KRR with i.i.d. data are reported in (Caponnetto & De Vito, 2007; Steinwart et al., 2009). The last term in the error bound highlights that the error induced by the proposed solver in (13) is an order-one scheme with an additional factor contributed by errors in the estimation of the tangent space. More work needs to be done for non-i.i.d. data, especially when the training data consists of time series of solutions.

## 4 Related Work

Learning dynamics with geometrical constraints has been an active research field for decades. Early studies considered constraining the high-dimensional dynamics in a linear subspace to obtain a low-dimensional model (Benner et al., 2015). However, such a linear subspace approach is not adequate to represent even simple manifolds such as the sphere and torus. Recent approaches in approximating vector fields on the tangent bundles can be categorized into two classes.

The first class either assumes the knowledge of the manifolds (Elamvazhuthi et al., 2023; Jeong et al., 2023) or employs an empirical procedure that relies on the domain knowledge to globally parameterize the manifolds (Loiseau et al., 2021; Callaham et al., 2022). In the latter approach, the dynamics are identified using SINDy with a special set of polynomials that are only appropriate for describing simple manifolds, such as the torus. Since our method shares a similar vein to this approach, which they called Manifold-SINDy, we provide a numerical comparison of their example, the cavity flow problem, in Section 5. We shall see that our method yields significantly more accurate short and long-term predictions of the dynamics.

The second class of methods addresses arbitrary manifold structure. In Crosskey & Maggioni (2017); Floryan & Graham (2022); Zeng et al. (2024), an atlas of charts of a manifold is learned from the data, and the dynamics are learned within each chart so that the dynamics are guaranteed to evolve on the manifold. Specifically, CANDyMan uses k-means algorithm to cluster the data, and employs an autoencoder to represent data clusters as an atlas of the charts and feed-forward neural networks to model the dynamics in the latent variable induced by the local chart. Lou et al. (2020) employed a similar strategy but learned the atlas and dynamics simultaneously through a neural ODE framework. The main drawback of this class of methods is the complicated model form, which involves not only the charts and local dynamics but also the transition maps among the charts. In Section 5, we will compare GMKRR with one of the methods in this class, the CANDyMan proposed in Floryan & Graham (2022), on several examples in their paper, including the beating and travelling waves in Kuramoto-Sivashinsky equations, and reaction-diffusion dynamics. We shall see how GMKRR outperforms CANDyMan in training efficiency and prediction accuracy.

## 5 NUMERICAL EXPERIMENTS

In this section, we numerically test the proposed approach on several high-dimensional problems.

### 5.1 A 2D CAVITY FLOW PROBLEM

The GMKRR algorithm is demonstrated on a cavity flow problem (Callaham et al., 2022) that have high ambient dimensions and admit quasi-periodic dynamics on a 2D torus. Through this example, the predictive accuracy and long-term numerical stability of GMKRR are benchmarked against conventional SINDy methods. Two SINDy-type methods are considered for comparison: (1) Vanilla SINDy, denoted simply "SINDy" below, and (2) "Manifold SINDy" (M-SINDy) that is proposed in Callaham et al. (2022) to account for the manifold constraint. A trajectory of 30000 steps is available, and all models are trained using the first 4000 steps. The details of the dataset and hyperparameters of the three models are provided in Appendix E.2.

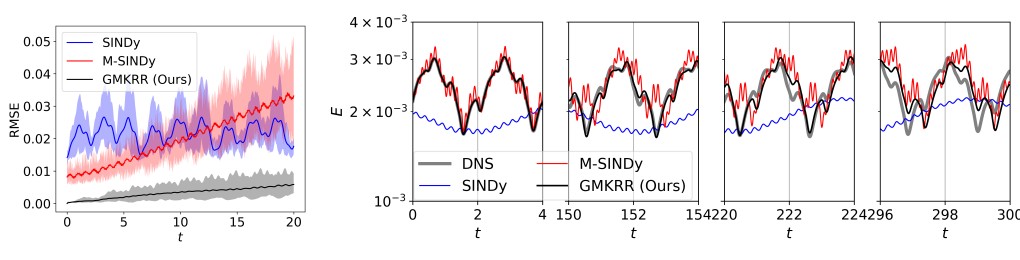

(a) Comparison of error growth.    (b) Comparison of kinetic energy in long-term prediction.

Figure 3: Results of 2D cavity flow.

The prediction errors of the three models are compared in Fig. 3a, where the solid lines and the shades show the evolution of the mean and range of RMSE over the test trajectories, respectively. The error in SINDy is large, and maintain at a similar level of error through the examined time horizon. The error in M-SINDy grows rapidly; this is attributed to the phase shift in the predicted dynamics. In addition, the M-SINDy predictions start with a non-negligible error at the beginning, this is due to the error of the manifold representation learned from data. As for GMKRR, the error maintains as low as 1/6 of the M-SINDy errors, with also a smaller standard deviation; such small error is granted from the enforcement of manifold constraint at every time step.

Next, the long-term numerical stability of the three methods is compared. In the cavity flow problem, the kinetic energy, computed as $E = \|\mathbf{x}\|^2$, fluctuates and maintains a constant amplitude and range, which indicates that the flow states stay on a low-dimensional manifold. Here, all the models are solved from the first data point for the entire 30000 time steps. The time evolution of $E$ is shown in Fig. 3b. The SINDy prediction gives completely wrong trend, which also indicates that it does not satisfy the manifold constraints. Both M-SINDy and GMKRR capture the fluctuation amplitudes well; particularly GMKRR follows the truth accurately up to $t = 150$ when M-SINDy already shows significant deviations. Furthermore, M-SINDy shows spurious high-frequency fluctuations throughout the simulation, that appears to be non-physical. Admittedly, GMKRR ends up with phase shift over long-term prediction, and this is arguably numerically unavoidable. However, the comparison above highlights the capability of GMKRR to produce accurate long-term predictions using a limited amount of data, outperforming SINDy and M-SINDy.

### 5.2 MORE HIGH-DIMENSIONAL PROBLEMS

Next, we benchmark GMKRR against the high-dimensional examples developed in CANDyMan (Floryan & Graham, 2022), that have well-defined manifold structures. These examples include the beating and travelling dynamics of the 1D Kuramoto-Sivashinsky (KS) equation and the 2D reaction-diffusion (RD) dynamics (Fig. 4(a-d)). For the KS travelling case, we also consider two additional baselines, NODE and LDNet, that admit low-dimensional (latent) structures but do not exactly account for intrinsic dimensions. The Isomap algorithm (Balasubramanian & Schwartz, 2002) is employed to visualize the intrinsic coordinates of these dynamics (Fig. 4(e-h)); note that

Isomap is only for visualization purposes and its results are not used in the subsequent model training and prediction. Furthermore, the comparison of prediction errors over time between GMKRR and the baselines is shown in Fig. 4(i-l). Finally, the computational costs and the error metrics of the models are provided in Table 1, where "RMSE" and "Max Error" refer to the average and maximum of RMSE along the trajectory. Throughout this section, the GMKRR model uses simple local SVD to estimate the tangent space, unless otherwise specified. More details on model hyperparameters, training, and result comparison are provided in Appendices E.3 and E.4.

**KS beating dynamics.** The dynamic oscillates at a single frequency and lie on a 1D submanifold in a 64-dimensional ambient space (Fig. 4(e)). This is a relatively easy case for both GMKRR and CANDyMan, but the error of GMKRR is one order of magnitude smaller than that of CANDyMan throughout the prediction horizon (Fig. 4(i)). Furthermore, the cost for training GMKRR is over 3000 times lower than CANDyMan.

**KS travelling dynamics.** The dynamic has two timescales that differ by a factor of approximately 200. The fast timescale corresponds to the oscillation of the spatial wave shape (Fig. 4(b)), while the slow timescale corresponds to the translation of the wave in space (Fig. 4(c)). The combination of the two timescales results in a 3D manifold (Fig. 4(f-g)). Learning the travelling dynamics is more challenging than in the previous case because the complete manifold cannot be covered without including at least a full period of the long timescale that spans a large number of data samples (approximately 5000 steps). Here, five models are considered: (1) CANDyMan: The model involves a special data preprocessing step based on prior knowledge (see Appendix E.3) so that the fast and slow timescale dynamics are learned separately, and only the first 100 steps are needed for training. (2) NODE: The modeling strategy and preprocessing procedure are identical to CANDyMan, except that the fast dynamic is represented by a generic NN without assuming any low-dimensional structure. (3) LDNet: The same setup as NODE, except that the fast dynamic is learned in a 2D latent space. (4) GMKRR-Full: The model does not consider any prior knowledge as in the CANDyMan, since such knowledge is not always available. (5) GMKRR-FFT: As an apple-to-apple comparison against CANDyMan, this model is trained using exactly the same data preprocessing strategy and dataset as CANDyMan.

In the initial period, which covers a few short timescales, the GMKRR models clearly outperform CANDyMan and LDNet and are comparable with NODE (Fig. 4(j)). Notice that LDNet also admits an encoding-decoding error with large RMSE at initial time. In the long term, the baselines NODE and LDNet, that do not account for intrinsic dimensions, perform the worst; this is not surprising as the geometrical constraints are not enforced during prediction. The error of GMKRR-Full is similar to CANDyMan, although the former does not leverage the prior knowledge like the latter. Lastly, the error of GMKRR-FFT is consistently 1-2 orders of magnitude smaller than all the other baselines (Fig. 4(k)). Furthermore, while GMKRR-FFT and the baselines are trained using the identical dataset, the former is 2-3 orders of magnitude faster than the latter. Clearly, GMKRR beats most baselines even without any prior knowledge, and considering the prior knowledge only expands the advantage of GMKRR over all the baselines.

**Reaction-diffusion dynamics.** The dynamics oscillates at a single frequency (Fig. 4(d)), and lives on a 1D submanifold (Fig. 4(h)). However, the data is 20402-dimensional and hence may pose a computational challenge to both GMKRR and CANDyMan. First we consider CANDyMan and "GMKRR-Full", which are directly trained using the original 20402-dimensional data. When compared to CANDyMan, GMKRR-Full achieves two-orders-of-magnitude lower error in prediction (Fig. 4(l)). While the training cost of GMKRR-Full is 1/72 of CANDyMan, the former does take longer time in prediction, due to the need for GMLS in normal correction in all 20402 dimensions. Next, we devise the "GMKRR-PCA" models to reduce the computational cost. The model first reduces the data by PCA into 6 dimensions, and then learn a GMKRR model in the PCA coordinates. In the prediction phase, the GMKRR first predicts the dynamics in the 6 PCA coordinates and then maps back to the original space by the PCA modes. The "GMKRR-PCA-T0" model uses the local SVD technique to estimate the tangent space, while the "GMKRR-PCA-T4" model uses a 4th-order estimation by GMLS. When compared to GMKRR-Full, the computational costs are significantly reduced in GMKRR-PCA-T0 while maintaining nearly identical predictions, hence in Fig. 4(l), the curves of GMKRR-Full and GMKRR-PCA-T0 overlap with each other. Lastly, GMKRR-PCA-T4 shows improvement over the other GMKRR models, with a mild trade-off between cost and error.

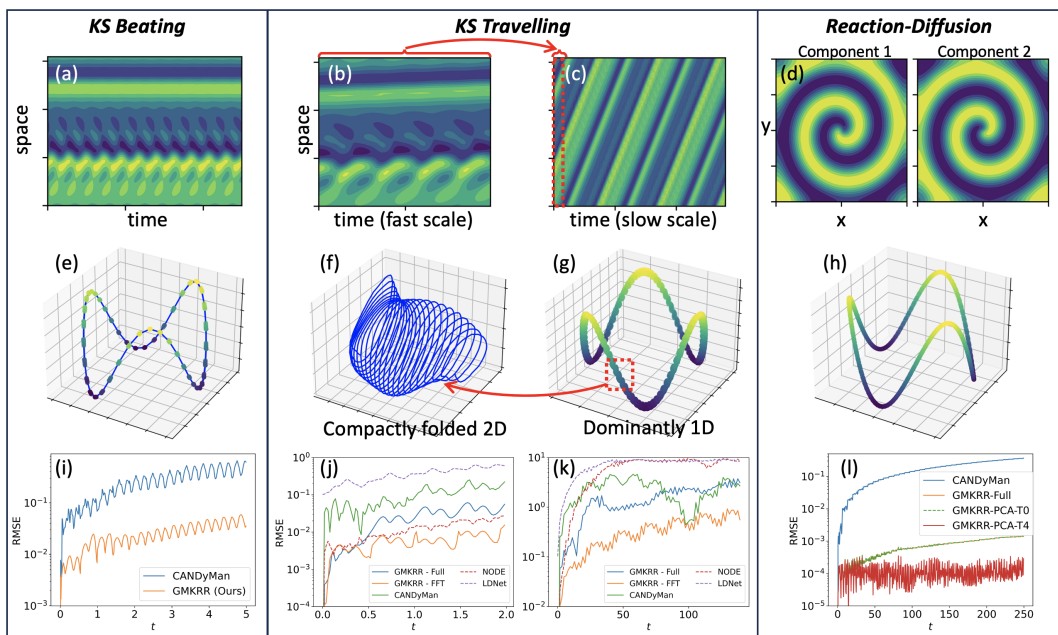

Figure 4: Comparison of GMKRR and CANDyMan: (a-d) data; (e-h) visualizations of low-dimensional manifold structure; (i-l) Prediction errors.

Table 1: Model performance for the CANDyMan cases

| CASE | MODEL | Training Time (s) | Prediction Time (s) | RMSE | Max Error |
|---|---|---|---|---|---|
| KS Beating | CANDyMan | 48.04 | 0.6703 | 0.2744 | 2.130 |
| | GMKRR | **0.01367** | **0.5133** | **0.02549** | **0.1972** |
| KS Travelling | CANDyMan | 70.10 | 19.44 | 2.391 | 15.22 |
| | NODE | 395.6 | **4.166** | 6.690 | 22.14 |
| | LDNet | 77.87 | 13.53 | 7.600 | 22.46 |
| | GMKRR-Full | 11.68 | 20.82 | 1.604 | 12.45 |
| | GMKRR-FFT | **0.04550** | 24.41 | **0.3671** | **3.088** |
| Reaction-Diffusion | CANDyMan | 265.8 | 14.59 | 0.1797 | 0.5517 |
| | GMKRR-Full | 3.723 | 175.4 | 7.394E-4 | 2.959E-3 |
| | GMKRR-PCA-T0 | **0.01953** | **4.786** | 7.399E-4 | 2.961E-3 |
| | GMKRR-PCA-T4 | 0.1187 | 8.5855 | **1.216E-4** | **1.339E-3** |

## 6 CONCLUSIONS

We have developed GMKRR, in tandem with a geometry-preserving time integrator, for learning dynamics on manifolds; it overcomes the limitations of existing approaches in terms of computational efficiency and prediction accuracy. The GMKRR involves an intrinsic kernel formulation, so that its training and prediction are highly scalable for data in high-dimensional ambient spaces. The empirical results show that our models outperform the classical SINDy-based and neural-network-based methods significantly, especially in the long-term prediction accuracy of dynamics on manifolds. We also contribute theoretically by providing the error bound of the prediction by the proposed methods. Future work may focus on improving the scalability to large datasets using, e.g., pseudo-input strategies, extending to dynamics with parameters and inputs, and extending the framework to combat noisy data, leveraging ideas from Candès et al. (2011); Messenger & Bortz (2021); Fasel et al. (2022).

REPRODUCIBILITY STATEMENT

We are committed to ensuring the reproducibility of our results. The code used to generate all the figures (except Fig. 2) and the results in Table 1 is available. At the stage of double-blind review, the code is provided as a ZIP file in Supplementary Material on OpenReview. Detailed instructions for running the experiments, including the configuration of hyperparameters and execution of the analysis, are provided in the README. The datasets used in our work are either publicly available or described with clear instructions for downloading and preprocessing in the repository. All steps required to replicate our experiments and analyses are documented to enable seamless reproduction by the reviewers and future researchers.

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

# A   A LOCAL PARAMETERIZATION OF SMOOTH MANIFOLDS BY GENERALIZED MOVING LEAST-SQUARES METHODS

In this appendix, we discuss a local SVD method to approximate local tangent space, review the Generalized Moving Least Squares (GMLS) method, and its application to approximate a local parameterization of smooth manifolds which in turn gives us a higher-order approximation of the local tangent space, and discuss an algorithm for the normal correction for high-dimensional ambient space.

Let $X = \{\mathbf{x}_i\}_{i=1}^N \subset M \subset \mathbb{R}^n$ be a set of point cloud data that is available for training. For any base point $\mathbf{x}_0 \in M$ ($\mathbf{x}_0$ can be either in the training data set $X$ or any new points), we denote its $K-$nearest neighbors in $X$ by $S_{\mathbf{x}_0} = \{\mathbf{x}_{0,k}\}_{k=1}^K \subset X$. By definition, we have $\mathbf{x}_{0,1} = \mathbf{x}_0$ to be the base point.

## A.1   LOCAL SVD APPROXIMATION TO THE LOCAL TANGENT SPACE AND ITS DIMENSION

The classical local SVD method Donoho & Grimes (2003); Zhang & Zha (2004) approximate $\mathbf{T_0}$ as follows:

1. Construct the distance matrix $\mathbf{D} := [\mathbf{D}_1, \ldots, \mathbf{D}_K] \in \mathbb{R}^{n \times K}$, where $K > d$ and $\mathbf{D}_i := \mathbf{x}_{0,i} - \mathbf{x}_0$.

2. Take a singular value decomposition of $\mathbf{D} = \boldsymbol{U}\boldsymbol{\Sigma}\mathbf{V}^\top$. Let $\tilde{\mathbf{T}}_0$ be the leading $d-$columns of $\mathbf{U}$, which approximates an orthonormal basis of $T_{\mathbf{x}_0}M$.

We should point out that this approximation has a convergence of order $N^{-1/d}$ for $K = d^4 n \log n |K_{max}|$, where $K_{max}$ denotes the maximum principal curvature and for uniformly sampled data set $X$ of size $N$ denotes the size of training data (See Remark 9 in Harlim et al. (2023)).

## A.2   GENERALIZED MOVING LEAST SQUARES

To attain a higher-order approximation, we consider the generalized moving least square method. Generally, consider the regression for a smooth function $f \in C^\infty(M)$. Denote $\mathbf{f}_{\mathbf{x}_0} = (f(\mathbf{x}_{0,1}), \ldots, f(\mathbf{x}_{0,K}))^\top$. We consider using the intrinsic polynomials to approximate smooth functions over a neighborhood of $\mathbf{x}_0$. By intrinsic, we refer to the polynomial defines on a local chart with base point $\mathbf{x}_0$.

First, let $\mathbb{P}_{\mathbf{x}_0}^{l,d}$ be the space of intrinsic polynomials with degree up to $l$ in $d$-dimensions at the point $\mathbf{x}_0$, i.e., $\mathbb{P}_{\mathbf{x}_0}^{l,d} = \text{span}(\{p_{\mathbf{x}_0,\boldsymbol{\alpha}}\}_{|\boldsymbol{\alpha}| \leq l})$, where $\boldsymbol{\alpha} = (\alpha_1, \ldots, \alpha_d)$ is the multi-index notation with $|\boldsymbol{\alpha}| = \alpha_1 + \ldots + \alpha_d$ and $p_{\mathbf{x}_0,\boldsymbol{\alpha}}$ is the basis polynomial functions defined as

$$p_{\mathbf{x}_0,\boldsymbol{\alpha}}(\mathbf{x}) = \prod_{i=1}^d \left[\tilde{\boldsymbol{t}}_i(\mathbf{x}_0) \cdot (\mathbf{x} - \mathbf{x}_0)\right]^{\alpha_i}, \quad |\boldsymbol{\alpha}| \leq l.$$

Here, $\{\tilde{\boldsymbol{t}}_i(\mathbf{x}_0)\}_{i=1}^d$ are the $d$ tangent vectors at the base point $\mathbf{x}_0$ that are estimated by the local SVD scheme, $\tilde{\mathbf{T}}_0 = (\tilde{\boldsymbol{t}}_1(\mathbf{x}_0), \ldots, \tilde{\boldsymbol{t}}_d(\mathbf{x}_0)))$. By definition, the dimension of the space $\mathbb{P}_{\mathbf{x}_0}^{l,d}$ is $m = \binom{l+d}{d}$.

For $K > m$, we can define an operator $\mathscr{I}_\mathbb{P} : \mathbf{f}_{\mathbf{x}_0} \in \mathbb{R}^K \to \mathscr{I}_\mathbb{P}\mathbf{f}_{\mathbf{x}_0} \in \mathbb{P}_{\mathbf{x}_0}^{l,d}$ such that $\mathscr{I}_\mathbb{P}\mathbf{f}_{\mathbf{x}_0}$ is the optimal solution of the following least-squares problem:

$$\min_{\hat{f} \in \mathbb{P}_{\mathbf{x}_0}^{l,d}} \sum_{k=1}^K \left(f(\mathbf{x}_{0,k}) - \hat{f}(\mathbf{x}_{0,k})\right)^2. \tag{14}$$

The solution to the least-squares problem (14) can be represented as

$$\mathscr{I}_\mathbb{P}\mathbf{f}_{\mathbf{x}_0} = \sum_{|\boldsymbol{\alpha}| \leq l} b_{\boldsymbol{\alpha}} p_{\mathbf{x}_0,\boldsymbol{\alpha}},$$

where the concatenated coefficients $\boldsymbol{b} = (b_{\boldsymbol{\alpha}(1)}, \ldots, b_{\boldsymbol{\alpha}(m)})^\top$ satisfy the normal equation,

$$(\boldsymbol{\Phi}^\top\boldsymbol{\Phi})\boldsymbol{b} = \boldsymbol{\Phi}^\top\mathbf{f}_{\mathbf{x}_0}. \tag{15}$$

Notice that here is $m$ number of coefficients $\boldsymbol{\alpha}$ such that $|\boldsymbol{\alpha}| \leq l$ and thereafter we used the notation $\{\boldsymbol{\alpha}(j)\}_{j=1,\dots,m}$ to denote all possible $m$ multi-indices. The above normal equation can be uniquely identified if $\boldsymbol{\Phi} \in \mathbb{R}^{K \times m}$, with components

$$\boldsymbol{\Phi}_{kj} = p_{\mathbf{x}_0, \boldsymbol{\alpha}(j)}(\mathbf{x}_{0,k}), \quad 1 \leq k \leq K, \; 1 \leq j \leq m, \tag{16}$$

is a full rank matrix.

For uniformly sampled data set $X$ of size $N$, it was shown that the convergence rate of the GMLS approximation with a polynomial of order-$l$ to the $\alpha$th derivative of $f$ is of order-$N^{-\frac{l+1-|\alpha|}{d}}$ (see Jiang et al., 2024). For the tangent space approximation, the function $f$ will be the local parameterization of the manifold (as discussed in the next section), and the tangent space will be its Jacobian such that $\alpha = 1$. Therefore, if we consider GMLS approximation with a polynomial of order $l > 1$, we effectively have a higher-order approximation to the local SVD method.

### A.3 GMLS APPROXIMATION TO THE LOCAL PARAMETERIZATION AND LOCAL TANGENT SPACE

In this section, we now describe the GMLS approximation to a local parameterization of the manifold $\iota : T_{x_0} M \equiv \mathbb{R}^d \to M \subset \mathbb{R}^n$ such that $\iota(0) = x_0$.

Given the global tangent vectors at the base point $\mathbf{x}_0$, $\{\tilde{\boldsymbol{t}}_i(\mathbf{x}_0)\}_{i=1,\dots,d}$, one can use the QR decomposition to approximate the normal basis vectors $\{\tilde{\boldsymbol{n}}_j(\mathbf{x}_0)\}_{j=1,\dots,n-d}$. Then one can define a local coordinate chart for the manifold near the base point using the embedding map $\iota$,

$$\iota_{\mathbf{x}_0}(\theta_1, \theta_2, \dots, \theta_d) = \mathbf{x}_0 + \theta_1 \tilde{\boldsymbol{t}}_1(\mathbf{x}_0) + \theta_2 \tilde{\boldsymbol{t}}_2(\mathbf{x}_0) + \dots + \theta_d \tilde{\boldsymbol{t}}_d(\mathbf{x}_0) + \sum_{j=1}^{n-d} z_j(\theta_1, \theta_2, \dots, \theta_d)\, \tilde{\boldsymbol{n}}_j(\mathbf{x}_0), \tag{17}$$

where $(\theta_1, \theta_2, \dots, \theta_d) \in \mathbb{R}^d$ with $\theta_j = \boldsymbol{t}_j(\mathbf{x}_0) \cdot (\mathbf{x} - \mathbf{x}_0)$ denote the local coordinates and $z_j = \boldsymbol{n}_j(\mathbf{x}_0) \cdot (\mathbf{x} - \mathbf{x}_0)$ is a smooth function over the $(\theta_1, \theta_2, \dots, \theta_d)$-plane. This local parametrization is known as the Monge-Gauge parametrization of the manifolds.

We will simply approximate $z_j : \mathbf{R}^d \to \mathbf{R}$ with the GMLS method as a function that maps the local coordinates $(\theta_1, \theta_2, \dots, \theta_d)$ to a point on the manifold $M$ by solving the local least squares problem:

$$\min_{\hat{z}_j \in \mathbb{P}_{\mathbf{x}_0}^{l,d}} \sum_{k=1}^{K} \left( z_{j,k} - \hat{z}_j(\mathbf{x}_{0,k}) \right)^2, \tag{18}$$

where the available data are,

$$z_{j,k} = \tilde{\boldsymbol{n}}_j(\mathbf{x}_0) \cdot (\mathbf{x}_{0,k} - \mathbf{x}_0).$$

In this notation, we are fitting a model of the following form (intrinsic polynomials),

$$\hat{z}_j(\theta_1, \dots, \theta_d) = \sum_{|\boldsymbol{\alpha}| \leq l} b_{j,\boldsymbol{\alpha}} \theta_1^{\alpha_1} \theta_2^{\alpha_2} \cdots \theta_2^{\alpha_d},$$

where $\hat{z}_j(\mathbf{x}_{0,k})$ is a shorthand notation for $\hat{z}_j(\theta_1^k, \dots, \theta_d^k)$ evaluated at the local coordinates of $\mathbf{x}_{0,k}$, i.e., $\theta_\ell^k = \tilde{\boldsymbol{t}}_\ell(\mathbf{x}_0) \cdot (\mathbf{x}_{0,k} - \mathbf{x}_0)$ for $\ell = 1, \dots, d$.

With this GMLS approximation, we also inherit the approximate tangent space from the columns of the Jacobian of $\hat{\iota}_{\mathbf{x}_0}$ defined as in (17) with $z_j$ replaced by its estimate $\hat{z}_j$,

$$D\hat{\iota}_{\mathbf{x}_0}(\theta_1, \theta_2, \dots, \theta_d) = \begin{pmatrix} \hat{\boldsymbol{t}}_1(\mathbf{x}_0) & \cdots & \hat{\boldsymbol{t}}_d(\mathbf{x}_0) \end{pmatrix}, \tag{19}$$

where

$$\hat{\boldsymbol{t}}_i(\mathbf{x}_0) = \tilde{\boldsymbol{t}}_i(\mathbf{x}_0) + \sum_{j=1}^{n-d} \frac{\partial}{\partial \theta_i} \hat{z}_j(\theta_1, \theta_2, \dots, \theta_d)\, \tilde{\boldsymbol{n}}_j(\mathbf{x}_0). \tag{20}$$

Effectively, the GMLS approximation improves the approximation (as discussed at the end of the previous subsection) by adding additional terms in the normal components in (20).

### A.4 GMLS for normal correction in high ambient dimensions

For problems of high ambient dimensions, when the above procedure is applied to the normal correction in the Euler+NC method, a computational bottleneck will emerge. That is, at every time step, all the $(n-d)$ normal vectors $\tilde{\boldsymbol{N}} = [\tilde{\boldsymbol{n}}_1, \tilde{\boldsymbol{n}}_2, \cdots, \tilde{\boldsymbol{n}}_{n-d}] \in \mathbb{R}^{n \times (n-d)}$ need to be computed to form the $(n-d)$ local least squares problems (18). The normal vectors may be obtained by, e.g., QR decomposition of $\tilde{\boldsymbol{T}} = [\tilde{\boldsymbol{t}}_1, \tilde{\boldsymbol{t}}_2, \cdots, \tilde{\boldsymbol{t}}_d] \in \mathbb{R}^{n \times d}$, which would cost $O(n(n-d)^2)$ and is intractable when $n$ is large. The same issue occurs in the regression problem (12), where one needs $\hat{\mathbf{N}}$ once $\hat{\mathbf{T}}$ is attained from the GMLS regression in (20).

To circumvent the explicit computation of the $\tilde{N}$, we employ the following projection-based approach. First we define the projection matrix $\boldsymbol{Q} = \tilde{\boldsymbol{N}}\tilde{\boldsymbol{N}}^\top$, which maps a point $\mathbf{x} \in \mathbb{R}^n$ to the subspace spanned by $\tilde{N}$. Equivalently $\boldsymbol{Q} = \boldsymbol{I} - \tilde{\boldsymbol{T}}\tilde{\boldsymbol{T}}^\top$, so that the formation of $\tilde{N}$ is avoided, and the matrix-vector product $\boldsymbol{Q}\mathbf{x} = \mathbf{x} - \tilde{\boldsymbol{T}}(\tilde{\boldsymbol{T}}^\top \mathbf{x})$ has a low cost of $O(dn)$. Next, the following modified least squares problem is defined:

$$\min_{\hat{x}_j \in \mathbb{Q}^{l,d}_{\mathbf{x}_0}} \sum_{k=1}^{K} \left\| \mathbf{y}_k - \hat{\mathbf{x}}(\mathbf{x}_{0,k}) \right\|^2, \tag{21}$$

where the data

$$\mathbf{y}_k = \boldsymbol{Q}(\mathbf{x}_{0,k} - \mathbf{x}_0) \in \mathbb{R}^n,$$

and the model $\hat{\mathbf{x}} = [\hat{x}_1, \hat{x}_2, \cdots, \hat{x}_n]$

$$\hat{\mathbf{x}}(\theta_1, \ldots, \theta_d) = \sum_{|\boldsymbol{\alpha}| \leq l} \mathbf{b}_{\boldsymbol{\alpha}} \theta_1^{\alpha_1} \theta_2^{\alpha_2} \cdots \theta_2^{\alpha_d}.$$

Problem (21) is essentially $n$ local least squares problems; once it is solved, the normal components in (17) are computed collectively as

$$\sum_{j=1}^{n-d} z_j (\theta_1, \theta_2, \ldots, \theta_d) \, \tilde{\boldsymbol{n}}_j(\mathbf{x}_0) = \boldsymbol{Q}\hat{\mathbf{x}}(\theta_1, \ldots, \theta_d).$$

In the case where the ambient dimension $n$ is extremely high, the solution of (21) can be parallelized to further accelerate the computation.

The same approach is used to avoid an explicit computation of $\hat{\mathbf{N}}$ by replacing $\tilde{\mathbf{T}}$ with $\hat{\mathbf{T}}$ in the formulation above.

## B  Estimation of the Intrinsic Dimension of a Manifold

A critical parameter in GMKRR is the intrinsic dimension $d$ of the manifold. Here two algorithms for estimating $d$ from data are provided.

### B.1  Local SVD

Following notation in Appendix A, let the diagonal components of $\boldsymbol{\Sigma}$, the singular values of $\mathbf{D}$, be denoted by $\sigma_1 \geq \sigma_2 \geq \cdots \geq \sigma_K$. To determine $d$, we normalize the singular values by defining,

$$\xi_i^2 = \frac{\sigma_i^2}{\sum_{i=1}^{k} \sigma_i^2}.$$

We choose $d_0 = d_0(\delta)$ as an approximation to the local tangent space $T_{\mathbf{x}_0} M$ that depends on a fixed threshold $\delta \in (0, 1)$ chosen such that,

$$d_0(\delta) = \arg\max_{d \geq 1} \left\{ d : \sum_{i \leq d} \xi_i^2 \leq \delta \right\}.$$

This threshold must be tuned to the data, noting that higher thresholds will yield higher dimensional tangent spaces and lower thresholds will yield lower dimensional tangent spaces with all the trade-offs that this implies.

### B.2 KERNEL-BASED METHOD

An alternative method for dimension estimation that avoids tuning is provided in Coifman et al. (2008). The method relies on an ad hoc choice of scalar-valued kernel $\rho(\mathbf{x}, \mathbf{x}'; \eta)$ with a bandwidth parameter $\eta$.

Given a dataset of point $\{\mathbf{x}_i\}_{i=1,2,\cdots,N}$, define the sum

$$S(\eta) = \frac{1}{N^2} \sum_{i,j=1}^{N} \rho(\mathbf{x}_i, \mathbf{x}_j; \eta)$$

One can verify that $\lim_{\eta \to 0} S(\eta) = 1/N$ and $\lim_{\eta \to \infty} S(\eta) = 1$. Coifman et al. (2008) shows that $S(\eta) \propto \eta^{d/2}$ in a range of $\eta$, and proposed the following estimation of intrinsic dimension. First define

$$\tilde{d}(\eta) = \frac{2 \, \mathrm{d} \log(S(\eta))}{\mathrm{d} \log(\eta)},$$

then the estimated intrinsic dimension is $d^* = \tilde{d}(\eta^*)$, where

$$\eta^* = \arg\max_{\eta} \tilde{d}(\eta)$$

In the numerical implementation of the above intrinsic dimension estimation method, the derivative $\tilde{d}(\eta)$ can be estimated analytically, since $\tilde{d}(\eta) = \frac{\eta}{S} / \frac{\mathrm{d}S}{\mathrm{d}\eta}$, or simply by finite difference. Furthermore, we suggest using a modified SE kernel $\rho(\mathbf{x}, \mathbf{x}'; \eta) = \exp\left(-\|\mathbf{x} - \mathbf{x}'\|^2 / (L^2 \eta)\right)$, where $L$ is the maximum of pair-wise L2 distances of the dataset; this modification normalizes the distances and makes the search for maximum $\tilde{d}$ more robust.

Finally, as a side note, while the above bandwidth parameter $\eta$ is developed for intrinsic dimension estimation, we found empirically that a good initial guess for the bandwidth parameter $\gamma$ in the SE kernel of GMKRR can be computed based on $\eta^*$,

$$\gamma^* = 10(L^2 \eta^*)^{1/d^*}.$$

In the KS and reaction-diffusion cases presented in this paper, to perform the cross-validation in GMKRR, we empirically set a grid search of, e.g., $0.8\gamma^*, 1.0\gamma^*, 1.2\gamma^*$, etc. Yet, it was found that choosing $\gamma = \gamma^*$ for the SE kernel is sufficient for accurate GMKRR predictions in some cases.

## C COMPLEXITY OF EACH PREDICTION STEP

To attain $\hat{\mathbf{T}}$ and $\hat{\mathbf{N}}$ in Step 1, we employ $K$−nearest neighbors (knn). A generic knn algorithm time complexity is $O(N(d + K))$, where $K$ denotes the number of nearest neighbors. Subsequently, we solve an SVD of a matrix of size $n \times K$, where $n$ denotes the dimension of ambient space, the time complexity of local SVD is $O(nK \max\{n, K\} + \min\{n, K\}^3)$ to obtain $\tilde{\mathbf{T}}$ and $\tilde{\mathbf{N}}$. Since we use the higher-order approximation in (20), GMLS regression is employed in parallel for each co-dimensional coordinate. In each coordinate, we solve a linear problem of size $m \times m$, where $m < K$ denotes the number of monomials in GMLS fitting, so the cost in constructing $\Phi^\top \Phi$ in (15) is $O(Km^2)$ and inverting is $O(m^3)$, which is cheaper than the SVD. So, the total cost is still dominated by local SVD. For problems with $n \gg K$, see Appendix A.4 for an efficient numerical implementation to avoid forming the normal vectors $\tilde{\mathbf{N}}$ in the SVD step and the improved normal vector estimates $\hat{\mathbf{N}}$ after we obtained $\hat{\mathbf{T}}$ from the GMLS step. The complexity of these operations is far less than the computations above as reported in Appendix A.4.

In Step 2 of the algorithm, we employ the GMLS again to attain each component of $\hat{g}$ in parallel. Each component takes $O(m^2 K + m^3)$ operations.

The cost of evaluating $\mathbf{f}_\epsilon(\mathbf{x}) = \hat{\mathbf{T}}_\mathbf{x} \mathbf{r}(\mathbf{x}) \boldsymbol{\alpha}$ includes the total computational costs of the first step above to attain $\hat{\mathbf{T}}_\mathbf{x}$, the matrix-matrix-vector multiplication, $O(Nd^2 + nd)$, the evaluation of $\mathbf{r}(\mathbf{x})$ in (9), which is $O(d^2 nN)$, and the cost of evaluating exponentiation in the SE and Matérn kernels.

The total cost of evaluating (13) consists of the total cost of evaluating $\mathbf{f}_\epsilon(\mathbf{x})$, the time complexity in multiplying $\hat{\mathbf{N}}\hat{g}$, which is $n(n - d)$, the $O(m^2 K + m^3)$ operations in GMLS fitting from Step 2 to

attain $\hat{g}$, and the complexity in evaluating $n - d$ component of $\hat{g}$, which is $(n-d)dm$, since we fit the GMLS $d$-dimensional model with a total of $m$ parameters in each coordinate. If we use the implicit approach in Appendix A.4, then the multiplication $\hat{\mathbf{N}}\hat{g}$ is implicitly done, and the total computation cost reduce from $n(n-d) + (n-d)dm$ to $ndm$.

# D PROOFS

## D.1 PROOF OF LEMMA 2.1

*Proof.* First, for any pair $(\mathbf{x}, \mathbf{x}') \in M \times M$, $k(\mathbf{x}, \mathbf{x}')^* = \rho(\mathbf{x}, \mathbf{x}')^* (\hat{P}(\mathbf{x})\hat{P}(\mathbf{x}'))^\top = \rho(\mathbf{x}', \mathbf{x})\hat{P}(\mathbf{x}')\hat{P}(\mathbf{x}) = k(\mathbf{x}', \mathbf{x})$. Second, for every finite set of points $\{\mathbf{x}_i\}_{i=1}^N$ in $M$ and $\{\mathbf{f}_i\}_{i=1}^N$ in $\mathfrak{X}(M)$,

$$\sum_{i,j=1}^N \langle \mathbf{f}_i, k(\mathbf{x}_i, \mathbf{x}_j)\mathbf{f}_j \rangle = \sum_{i,j=1}^N \mathbf{f}_i^\top \rho(\mathbf{x}_i, \mathbf{x}_j)\hat{P}(\mathbf{x}_i)\hat{P}(\mathbf{x}_j)\mathbf{f}_j$$

$$= \sum_{i,j=1}^N \rho(\mathbf{x}_i, \mathbf{x}_j)\mathbf{u}_i^\top \mathbf{u}_j = \sum_{i,j=1}^N \rho(\mathbf{x}_i, \mathbf{x}_j) \sum_{k=1}^n u_{ik} u_{jk}$$

$$= \sum_{k=1}^n \sum_{i,j=1}^N \rho(\mathbf{x}_i, \mathbf{x}_j) u_{ik} u_{jk} \geq 0$$

where $\mathbf{u}_i = \hat{P}(\mathbf{x}_i)\mathbf{f}_i \equiv [u_{i1}, u_{i2}, \cdots, u_{in}]^\top$, and the last row is because that $\rho$ is positive semi-definite so each term in the summation over $k$ is non-negative. Hence $k$ is an operator-valued kernel. $\square$

## D.2 PROOF OF THEOREM 3.1

Consider the following initial value problem involving a system of ODEs,

$$\dot{\mathbf{x}}(t) = \mathbf{f}(\mathbf{x}(t)), \quad t > t_0, \quad \mathbf{x}(t_0) = \mathbf{x}_0, \tag{22}$$

where $t, t_0 \in I \subset \mathbb{R}$, $\mathbf{x} \in C^1(I)$ and $\mathbf{f} : M \subset \mathbb{R}^n \to TM \subset \mathbb{R}^n$ is Lipschitz continuous. For the discussion below, we assume that (22) has an invariant measure $\mu$.

**Assumption D.1.** *Let $\mathbf{f}_\epsilon$ be an approximation of $\mathbf{f}$, such that $\|\mathbf{f} - \mathbf{f}_\epsilon\|_{L^2(\mu)} = O(\epsilon)$ as $\epsilon \to 0$. We also denote $\mathbf{x}_\epsilon$ that solves the approximate dynamics,*

$$\dot{\mathbf{x}}_\epsilon(t) = \mathbf{f}_\epsilon(\mathbf{x}_\epsilon(t)), \quad t > t_0, \quad \mathbf{x}_\epsilon(t_0) = \mathbf{x}_0. \tag{23}$$

*Here, $\mathbf{f}_\epsilon$ is assumed to be Lipschitz continuous.*

To facilitate the analysis below, we review the following basic result:

**Lemma D.1** (Gronwall). *Let $p \in L^1([0, T])$, that is, $\int_0^T p(t)\,dt < \infty$. Also let $p \geq 0$ and $g, \varphi \in C([0, T])$ with $g$ nondecreasing. If*

$$\varphi(t) \leq g(t) + \int_0^t p(s)\varphi(s)\,ds,$$

*then*

$$\varphi(t) \leq g(t) \exp\left(\int_0^t p(s)\,ds\right), \quad \forall t \in [0, T].$$

Let's first state the error induced by the approximate vector field $\mathbf{f}_\epsilon$. In the following, the expectation is defined with respect to the distribution of the initial conditions, $\mathbf{x}_0 \sim \mu$. Particularly, the expectation of the random variable $\mathbf{x}(t)$ is understood as integration over the distribution of $\mathbf{x}(t)$ that solves (22) with initial distribution $\mathbf{x}_0 \sim \mu$.

**Proposition D.1.** *Let the Assumption D.1 be valid. For any $T > 0$,*

$$\mathbb{E}_\mu \left[ \|\mathbf{x}(T) - \mathbf{x}_\epsilon(T)\|^2 \right] \leq C(T)\epsilon^2,$$

*where $C(T) = ce^{2LT}T^2 > 0$, for some constant $c > 0$ independent of $T$.*

*Proof.* For any fixed $t > 0$, Then,

$$
\begin{aligned}
\|\mathbf{x}(t) - \mathbf{x}_\epsilon(t)\| &\leq \int_0^t \|\mathbf{f}(\mathbf{x}(s)) - \mathbf{f}_\epsilon(\mathbf{x}_\epsilon(s))\| \, ds \\
&\leq \int_0^t \|\mathbf{f}(\mathbf{x}(s)) - \mathbf{f}_\epsilon(\mathbf{x}(s))\| \, ds + \int_0^t \|\mathbf{f}_\epsilon(\mathbf{x}(s)) - \mathbf{f}_\epsilon(\mathbf{x}_\epsilon(s))\| \, ds \\
&\leq \int_0^t \|\mathbf{f}(\mathbf{x}(s)) - \mathbf{f}_\epsilon(\mathbf{x}(s))\| \, ds + L \int_0^t \|\mathbf{x}(s) - \mathbf{x}_\epsilon(s)\| \, ds.
\end{aligned}
$$

where $L > 0$ is the Lipschitz constant for $\mathbf{f}_\epsilon$. By Gronwall, we have,

$$
\|\mathbf{x}(t) - \mathbf{x}_\epsilon(t)\| \leq e^{Lt} \int_0^t \|\mathbf{f}(\mathbf{x}(s)) - \mathbf{f}_\epsilon(\mathbf{x}(s))\| \, ds.
$$

By the Jensen's inequality,

$$
\|\mathbf{x}(t) - \mathbf{x}_\epsilon(t)\|^2 \leq e^{2Lt} t^2 \left( \frac{1}{t} \int_0^t \|\mathbf{f}(\mathbf{x}(s)) - \mathbf{f}_\epsilon(\mathbf{x}(s))\| \, ds \right)^2 \leq e^{2Lt} t \int_0^t \|\mathbf{f}(\mathbf{x}(s)) - \mathbf{f}_\epsilon(\mathbf{x}(s))\|^2 \, ds.
$$

Taking expectation, applying Fubini's theorem and the assumption,

$$
\mathbb{E}_\mu \left[ \|\mathbf{x}(t) - \mathbf{x}_\epsilon(t)\|^2 \right] \leq e^{2Lt} t \int_0^t \mathbb{E}_\mu \left[ \|\mathbf{f}(\mathbf{x}(s)) - \mathbf{f}_\epsilon(\mathbf{x}(s))\|^2 \right] \, ds \leq c e^{2Lt} t^2 \epsilon^2,
$$

for some $c > 0$. $\qquad \square$

For convenience of the discussion, let us state the GMLS error bound in terms of the fill distance Mirzaei et al. (2012),

$$
h_{X,M} = \sup_{x \in M} \min_{\mathbf{x}_i \in X} d_g(\mathbf{x}, \mathbf{x}_i),
$$

where $d_g : M \times M \to \mathbb{R}^+$ denotes the geodesic distance.

**Lemma D.2.** *Let $X = \{\mathbf{x}_1, \ldots, \mathbf{x}_N\} \subset M$ be a quasi-uniformly distributed data with $h_{X,M} \leq h_0$. Define $M^* = \bigcup_{\mathbf{x} \in M} B(\mathbf{x}, C_2 h_0)$, a union of geodesic ball over the length $C_2 h_0$, for some constant $C_2 > 0$. Then for any real-valued function $f \in C^{\ell+1}(M^*)$ and $\alpha$ that satisfies $|\alpha| \leq \ell$, with probability higher than $1 - \frac{1}{N}$,*

$$
\left| D^\alpha f(\mathbf{x}) - \widehat{D^\alpha f}(\mathbf{x}) \right| \leq c h_{X,M}^{\ell+1-|\alpha|} |f|_{C^{\ell+1}(M^*)}, \tag{24}
$$

*for all $\mathbf{x} \in M$ and some $c > 0$. Here the semi-norm*

$$
|f|_{C^{\ell+1}(M^*)} := \max_{|\beta| = \ell+1} \|D^\beta f\|_{L^\infty(M^*)},
$$

*is defined over $M^*$. In the error bound above, we used the notation $\widehat{D^\alpha f}$ as the GMLS approximation to $D^\alpha f$ using local polynomials up to degree $\ell$, where $D^\alpha$ denotes a general multi-dimensional derivative with multiindex $\alpha$.*

**Assumption D.2.** *Let the training data set $X$ be sampled i.i.d. with sampling distribution $\mu$, which is absolutely continuous with respect to the volume measure, $d\mu = q \, d\text{Vol}$, with $0 < q_{min} \leq q \leq q_{max}$.*

Then we can show that:

**Lemma D.3.** *Let the Assumption D.2 be valid, then the filled distance*

$$
\mathbb{P}_{X \sim \mu}(h_{X,M} > \delta) \leq \exp\left(-CN\delta^d\right),
$$

*where $C = C(d)/Vol(M)$ is some constant that depends on the intrinsic dimension of the manifold $M$.*

*Proof.* Suppose $h_{X,M} > \delta$, so there is $\mathbf{x} \in M$ such that $\min_{\mathbf{x}_i \in X} d_g(\mathbf{x}, \mathbf{x}_i) \geq \delta$. So $B_\delta(\mathbf{x}) \cap X = \emptyset$. In other words, each $\mathbf{x}_i \in X$ is in $M \backslash B_\delta(\mathbf{x})$. This has measure $1 - \frac{\mu(B_\delta(\mathbf{x}))}{\mu(M)} \leq 1 - C\delta^d$, where the inequality is based on the lower bound estimate for the Volume of geodesic ball (Proposition 14 in Croke (1980)). Hence, $\min_{\mathbf{x}_i \in X} d_g(\mathbf{x}, \mathbf{x}_i) \geq \delta$ occurs with probability less than $(1 - C\delta^d)^N \leq \exp(-CN\delta^d)$. This completes the proof. $\qquad \square$

With this estimate, we conclude that with probability higher than $1 - \frac{1}{N}$, the fill distance is bounded from above,

$$h_{X,M} \leq \left( \frac{\text{Vol}(M)}{C(d)} \frac{\log(N)}{N} \right)^{\frac{1}{d}}. \tag{25}$$

The result above requires the sample data to be independent and identically distributed (i.i.d.). While in general the data may not be i.i.d. as they are time series, the training data for the 1D example are drawn i.i.d. from the sampling distribution $\mu$ (see Appendix E.1).

Using the same argument and repeating the proof of Lemma A.2 in Yan et al. (2023), one can show that with probability higher than $1 - \frac{2}{N}$, the separation distance,

$$c_1(d) \frac{q_{min}}{q_{max}} N^{-2/d} \leq q_{X,M} := \frac{1}{2} \min_{i \neq j} d_g(\mathbf{x}_i, \mathbf{x}_j) \leq C_2(d) \left( \frac{\log N}{N^2} \right)^{1/d},$$

for some $c_1(d), C_2(d) 0$), so the quasi-uniformity assumption in Lemma (D.2) for a fixed $X$ of size $N$,

$$q_{X,M} \leq h_{X,M} \leq c_q q_{X,M},$$

remains valid although with constant $c_q$ that grow on the order of $N^{1/d}$. So, we conclude that under the Assumption D.2, the GMLS error estimate for any $\mathbf{f} \in C^{\ell+1}(M^*)$ is given by,

$$\left\| D^\alpha \mathbf{f}(\mathbf{x}) - \widehat{D^\alpha \mathbf{f}}(\mathbf{x}) \right\|_{C^\infty} \leq \hat{C} \left( \frac{\log(N)}{N} \right)^{\frac{\ell+1-|\alpha|}{d}} |\mathbf{f}|_{C^{\ell+1}(M^*)}, \tag{26}$$

for $|\alpha| \leq l$ and a constant $\hat{C} = c \left( \frac{\text{Vol}(M)}{C(d)} \right)^{1/d}$ that depends on $d$ and $\text{Vol}(M)$ with probability higher than $1 - \frac{1}{N}$.

We now solve the ODE in (23) with the following approximation (which is exactly (13)),

$$\hat{\mathbf{x}}_{\epsilon,i+1} = \hat{\mathbf{x}}_{\epsilon,i} + \Delta t \mathbf{f}_\epsilon(\hat{\mathbf{x}}_{\epsilon,i}) + (\Delta t)^2 \hat{\mathbf{N}}_{\epsilon,i} \hat{g}(\hat{\mathbf{T}}_{\epsilon,i}^\top \mathbf{f}_\epsilon(\hat{\mathbf{x}}_{\epsilon,i})), \tag{27}$$

where we have defined $\hat{\mathbf{x}}_{\epsilon,i} := \hat{\mathbf{x}}_\epsilon(t_i)$.

Let $\gamma_i : [0,1] \to M \subset \mathbb{R}^n$ denotes the geodesic that satisfies $\gamma_i(0) = \mathbf{x}_{\epsilon,i} := \mathbf{x}_\epsilon(t_i)$ and $\gamma_i'(0) = \mathbf{f}_\epsilon(\mathbf{x}_{\epsilon,i})$. In the discussion below, we denote the true tangent and normal vectors at $\mathbf{x}_{\epsilon,i}$ by $\mathbf{T}_{\epsilon,i}$ and $\mathbf{N}_{\epsilon,i}$, respectively. Since $\exp_{\mathbf{x}_{\epsilon,i}}(\mathbf{f}_\epsilon(\mathbf{x}_{\epsilon,i})\Delta t) = \gamma_i(\Delta t)$, we can employ the following Taylor's expansion on the exponential map,

$$\begin{aligned} \mathbf{x}_{\epsilon,i+1} &= \exp_{\mathbf{x}_0}(\mathbf{f}_\epsilon(\mathbf{x}_{\epsilon,i})\Delta t) = \gamma_i(\Delta t) \\ &= \gamma_i(0) + \gamma_i'(0)\Delta t + \frac{1}{2}\gamma_i''(0)(\Delta t)^2 + O((\Delta t)^3) \\ &= \mathbf{x}_{\epsilon,i} + \mathbf{f}_\epsilon(\mathbf{x}_{\epsilon,i})\Delta t + \frac{1}{2}\alpha\left(\mathbf{f}_\epsilon(\mathbf{x}_{\epsilon,i}), \mathbf{f}_\epsilon(\mathbf{x}_{\epsilon,i})\right)(\Delta t)^2 + O((\Delta t)^3), \end{aligned} \tag{28}$$

where $\alpha$ is the second fundamental form of $M$, which is a quadratic function of $\mathbf{f}_\epsilon(\mathbf{x}_{\epsilon,i})$, that is,

$$\alpha\left(\mathbf{f}_\epsilon(\mathbf{x}_{\epsilon,i}), \mathbf{f}_\epsilon(\mathbf{x}_{\epsilon,i})\right) = \mathbf{N}_{\epsilon,i} \mathbf{N}_{\epsilon,i}^\top \nabla_{\overline{\mathbf{f}_\epsilon(\mathbf{x}_{\epsilon,i})}} \overline{\mathbf{f}_\epsilon(\mathbf{x}_{\epsilon,i})}, \tag{29}$$

where $\nabla_u v$ denotes the Euclidean $\mathbb{R}^n$ directional derivative of $v$ in the direction of $u$, and $\overline{\mathbf{f}_\epsilon(\mathbf{x}_{\epsilon,i})}$ is an extension of $\mathbf{f}_\epsilon(\mathbf{x}_{\epsilon,i})$, such that $\overline{\mathbf{f}_\epsilon(\mathbf{x}_{\epsilon,i})}|_M = \mathbf{f}_\epsilon(\mathbf{x}_{\epsilon,i})$.

On the other hand, it is clear that,

$$\mathbf{x}_{\epsilon,i+1} - \mathbf{x}_{\epsilon,i} = \mathbf{T}_{\epsilon,i} \mathbf{T}_{\epsilon,i}^\top(\mathbf{x}_{\epsilon,i+1} - \mathbf{x}_{\epsilon,i}) + \mathbf{N}_{\epsilon,i} \mathbf{N}_{\epsilon,i}^\top(\mathbf{x}_{\epsilon,i+1} - \mathbf{x}_{\epsilon,i}) = \mathbf{f}_\epsilon(\mathbf{x}_{\epsilon,i})\Delta t + \mathbf{N}_{\epsilon,i} \mathbf{N}_{\epsilon,i}^\top(\mathbf{x}_{\epsilon,i+1} - \mathbf{x}_{\epsilon,i}). \tag{30}$$

Subtracting (30) from (28) and using (29), and taking $\mathbf{N}_{\epsilon,i}^\top$, replacing $t$ with $\Delta t$, we have,

$$g(\mathbf{T}_{\epsilon,i}^\top \mathbf{f}_\epsilon(\mathbf{x}_{\epsilon,i}))(\Delta t)^2 := \mathbf{N}_{\epsilon,i}^\top(\mathbf{x}_{\epsilon,i+1} - \mathbf{x}_{\epsilon,i}) = \frac{(\Delta t)^2}{2} \mathbf{N}_{\epsilon,i}^\top \nabla_{\overline{\mathbf{f}_\epsilon(\mathbf{x}_{\epsilon,i})}} \overline{\mathbf{f}_\epsilon(\mathbf{x}_{\epsilon,i})} + O\left((\Delta t)^3\right), \tag{31}$$

which motivates the GMLS estimation with polynomial of degree $\ell \geq 2$.

In general, however, we do not have the underlying $\mathbf{T}_{\epsilon,i}$ and $\mathbf{N}_{\epsilon,i}$. If we numerically employ the GMLS to approximate the local parameterization $\iota : \mathbb{R}^d \to M$ with polynomial of degree-$s$ as prescribed in Appendix A.3, we have the error bound in (26) with $|\alpha| = 1$. That is, with probability higher than $1 - \frac{1}{N}$

$$\left\| \mathbf{T}_{\epsilon,i} - \widehat{\mathbf{T}}_{\epsilon,i} \right\|_\infty \le \hat{C} \left( \frac{\log(N)}{N} \right)^{\frac{s}{d}} |\iota|_{C^{s+1}(M^*)} := \delta,$$

where we have defined $\delta$ to simplify the notation. Using the Gram-Schmidt formula and basic norm triangle inequalities, one can deduce that the normal vector estimation has the same error bound, $\left\| \mathbf{N}_{\epsilon,i} - \widehat{\mathbf{N}}_{\epsilon,i} \right\|_\infty \le \delta$. This means that,

$$\alpha \left( \mathbf{f}_\epsilon(\mathbf{x}_{\epsilon,i}), \mathbf{f}_\epsilon(\mathbf{x}_{\epsilon,i}) \right) = \mathbf{N}_{\epsilon,i} \mathbf{N}_{\epsilon,i}^\top \nabla_{\overline{\mathbf{f}_\epsilon(\mathbf{x}_{\epsilon,i})}} \overline{\mathbf{f}_\epsilon(\mathbf{x}_{\epsilon,i})} = \widehat{\mathbf{N}}_{\epsilon,i} \widehat{\mathbf{N}}_{\epsilon,i}^\top \nabla_{\overline{\mathbf{f}_\epsilon(\mathbf{x}_{\epsilon,i})}} \overline{\mathbf{f}_\epsilon(\mathbf{x}_{\epsilon,i})} + O(\delta), \tag{32}$$

in the uniform sense. Also,

$$\begin{aligned}
\mathbf{x}_{\epsilon,i+1} - \mathbf{x}_{\epsilon,i} &= \widehat{\mathbf{T}}_{\epsilon,i} \widehat{\mathbf{T}}_{\epsilon,i}^\top (\mathbf{x}_{\epsilon,i+1} - \mathbf{x}_{\epsilon,i}) + \widehat{\mathbf{N}}_{\epsilon,i} \widehat{\mathbf{N}}_{\epsilon,i}^\top (\mathbf{x}_{\epsilon,i+1} - \mathbf{x}_{\epsilon,i}) \\
&= \mathbf{f}_\epsilon(\mathbf{x}_{\epsilon,i}) \Delta t + \widehat{\mathbf{N}}_{\epsilon,i} \widehat{\mathbf{N}}_{\epsilon,i}^\top (\mathbf{x}_{\epsilon,i+1} - \mathbf{x}_{\epsilon,i}) + \underbrace{\left( \widehat{\mathbf{T}}_{\epsilon,i} \widehat{\mathbf{T}}_{\epsilon,i}^\top - \mathbf{T}_{\epsilon,i} \mathbf{T}_{\epsilon,i}^\top \right) (\mathbf{x}_{\epsilon,i+1} - \mathbf{x}_{\epsilon,i})}_{O(\delta) \Delta t \| \mathbf{f}_\epsilon(\mathbf{x}_i) \|_\infty}. 
\end{aligned} \tag{33}$$

Subtracting (33) from (28), using (32), and taking $\widehat{\mathbf{N}}_{\epsilon,i}^\top$, replacing $t$ with $\Delta t$, we will numerically estimate the following quantity,

$$g(\widehat{\mathbf{T}}_{\epsilon,i}^\top \mathbf{f}_\epsilon(\mathbf{x}_{\epsilon,i}))(\Delta t)^2 := \widehat{\mathbf{N}}_{\epsilon,i}^\top (\mathbf{x}_{\epsilon,i+1} - \mathbf{x}_{\epsilon,i}) = \frac{(\Delta t)^2}{2} \widehat{\mathbf{N}}_{\epsilon,i}^\top \nabla_{\overline{\mathbf{f}_\epsilon(\mathbf{x}_{\epsilon,i})}} \overline{\mathbf{f}_\epsilon(\mathbf{x}_{\epsilon,i})} + O\left( (\Delta t)^3 + \delta \Delta t \right). \tag{34}$$

Based on the GMLS error bound in (26), we immediately have the following result.

**Lemma D.4.** *Let the Assumption D.2 be valid. Suppose that the solution generated by the system of ODEs in* (22) *lie on a smooth manifold, of at least $C^3$ such that* (28) *is valid. Let $\iota$ be at least $C^{s+1}(M^*)$, where $s \ge 2$. Furthermore, let $\hat{g}$ be the GMLS polynomial of order $\ell$ estimate for $g \in C^{\ell+1}(M^*)$ in* (34) *with $\ell \ge 2$. Then, with probability higher than $1 - \frac{2}{N}$,*

$$\begin{aligned}
\left\| g(\mathbf{T}_{\epsilon,i}^\top \mathbf{f}_\epsilon(\mathbf{x}_{\epsilon,i})) - \hat{g}(\widehat{\mathbf{T}}_{\epsilon,i}^\top \mathbf{f}_\epsilon(\mathbf{x}_{\epsilon,i})) \right\|_{C^\infty(\mathbb{R}^{n-d})} &\le \hat{C}_g \left( \frac{\log(N)}{N} \right)^{\frac{\ell+1}{d}} |g|_{C^{\ell+1}(M^*)} \\
&\quad + \hat{C}_\iota \Delta t \left( \frac{\log(N)}{N} \right)^{\frac{s}{d}} |\iota|_{C^{s+1}(M^*)} \| \mathbf{f}_\epsilon(\mathbf{x}_{\epsilon,i}) \|_\infty,
\end{aligned}$$

*where $\hat{C}_g$ and $\hat{C}_\iota$ are the two constants of the error bounds of the GMLS estimates for, respectively, $g$ and $\iota$, as defined in* (26).

*Proof.* By GMLS error estimate, it is clear that, by probability greater than $1 - \frac{1}{N}$,

$$\left\| g(\widehat{\mathbf{T}}_{\epsilon,i}^\top \mathbf{f}_\epsilon(\mathbf{x}_{\epsilon,i})) - \hat{g}(\widehat{\mathbf{T}}_{\epsilon,i}^\top \mathbf{f}_\epsilon(\mathbf{x}_{\epsilon,i})) \right\|_{C^\infty(\mathbb{R}^{n-d})} \le \hat{C}_g \left( \frac{\log(N)}{N} \right)^{\frac{\ell+1}{d}} |g|_{C^{\ell+1}(M^*)}$$

On the other hand, with probability greater then $1 - \frac{1}{N}$,

$$\begin{aligned}
\left\| g(\mathbf{T}_{\epsilon,i}^\top \mathbf{f}_\epsilon(\mathbf{x}_{\epsilon,i})) - g(\widehat{\mathbf{T}}_{\epsilon,i}^\top \mathbf{f}_\epsilon(\mathbf{x}_{\epsilon,i})) \right\|_{C^\infty(\mathbb{R}^{n-d})} &= \left\| (\mathbf{N}_{\epsilon,i} - \widehat{\mathbf{N}}_{\epsilon,i})^\top (\mathbf{x}_{\epsilon,i+1} - \mathbf{x}_{\epsilon,i}) \right\|_\infty \\
&\le \left\| \mathbf{N}_{\epsilon,i} - \widehat{\mathbf{N}}_{\epsilon,i} \right\|_\infty \left\| \mathbf{x}_{\epsilon,i+1} - \mathbf{x}_{\epsilon,i} \right\|_\infty \\
&\le \hat{C}_\iota \left( \frac{\log(N)}{N} \right)^{\frac{s}{d}} |\iota|_{C^{s+1}(M^*)} \Delta t \| \mathbf{f}_\epsilon(\mathbf{x}_{\epsilon,i}) \|_\infty, \tag{35}
\end{aligned}$$

where we have used the bound in (26) and the first order Taylor expansion in (28). Combining these two bounds, the proof is complete. $\square$

**Proposition D.2.** *Let $\mathbf{x}_\epsilon$ be the solution of* (23) *with Lipschitz $\mathbf{f}_\epsilon$. We approximate $\mathbf{x}_\epsilon$ numerically through the solver defined in* (27) *with $\hat{g}$ in* (34) *to be the GMLS approximant of $g$ as defined in* (31). *Let the assumption in Lemma D.4 be valid. Then, with probability higher than $1 - \frac{3}{N}$,*

$$\mathbb{E}_\mu \left[ \| \mathbf{x}_{\epsilon,i} - \hat{\mathbf{x}}_{\epsilon,i} \|^2 \right] \le \check{C} \Delta t^2 \left( \frac{\log(N)}{N} \right)^{\frac{2\min\left\{ \ell+1, s \right\}}{d}},$$

*as* $N \to \infty$ *and* $\Delta t \to 0$, *where* $\check{C}$ *depends on* $q_{max}$, $Vol(M)$, $T, L$, $|g|_{C^{\ell+1}(M^*)}$, $|\iota|_{C^{s+1}(M^*)}$, $\|\mathbf{f}_\epsilon(\mathbf{x}_{\epsilon,i})\|_\infty$, $\|\hat{g}\|_{C(M^*)}$.

*Proof.* To establish the convergence, let us state the local truncation error by inserting the true solution of (23) to the discrete solver in (27),

$$\tau_i \quad := \quad \frac{\mathbf{x}_{\epsilon,i+1} - \mathbf{x}_{\epsilon,i}}{\Delta t} - \mathbf{f}_\epsilon(\mathbf{x}_{\epsilon,i}) - \Delta t \hat{\mathbf{N}}_{\epsilon,i} \hat{g}(\hat{\mathbf{T}}_{\epsilon,i}^\top \mathbf{f}_\epsilon(\mathbf{x}_{\epsilon,i})),$$

$$= \quad \Delta t \left( \mathbf{N}_{\epsilon,i} g(\mathbf{T}_{\epsilon,i}^\top \mathbf{f}_\epsilon(\mathbf{x}_{\epsilon,i})) - \hat{\mathbf{N}}_{\epsilon,i} \hat{g}(\hat{\mathbf{T}}_{\epsilon,i}^\top \mathbf{f}_\epsilon(\mathbf{x}_{\epsilon,i})) \right), \tag{36}$$

where we have used (28) with quadratic and higher order terms expressed in terms of $g$ using (29) and (34). We first note that this scheme is indeed consistent, that is, with probability higher than $1 - \frac{3}{N}$

$$\|\tau_i\|_\infty \leq \Delta t \left( \left\| \mathbf{N}_{\epsilon,i} g(\mathbf{T}_{\epsilon,i}^\top \mathbf{f}_\epsilon(\mathbf{x}_{\epsilon,i})) - \mathbf{N}_{\epsilon,i} \hat{g}(\hat{\mathbf{T}}_{\epsilon,i}^\top \mathbf{f}_\epsilon(\mathbf{x}_{\epsilon,i})) \right\| + \left\| \mathbf{N}_{\epsilon,i} \hat{g}(\hat{\mathbf{T}}_{\epsilon,i}^\top \mathbf{f}_\epsilon(\mathbf{x}_{\epsilon,i})) - \hat{\mathbf{N}}_{\epsilon,i} \hat{g}(\hat{\mathbf{T}}_{\epsilon,i}^\top \mathbf{f}_\epsilon(\mathbf{x}_{\epsilon,i})) \right\| \right)$$

$$\leq \Delta t \left( \hat{C}_g \left( \frac{\log(N)}{N} \right)^{\frac{\ell+1}{d}} |g|_{C^{\ell+1}(M^*)} \right.$$

$$\left. + \hat{C}_\iota \left( \frac{\log(N)}{N} \right)^{\frac{s}{d}} |\iota|_{C^{s+1}(M^*)} \left( \|\mathbf{f}_\epsilon(\mathbf{x}_{\epsilon,i})\|_\infty \Delta t + \|\hat{g}\|_{C(M^*)} \right) \right),$$

where we have used the bound in (D.4) and the fact that $\left\| \mathbf{N}_{\epsilon,i} - \hat{\mathbf{N}}_{\epsilon,i} \right\|_\infty \leq \delta$ with probability $1 - \frac{1}{N}$. This bound effectively means that $\|\tau_i\|_\infty = O(\Delta t)$ as $\Delta t \to 0$, with constant stated as above.

Following the standard convergence analysis by subtracting (27) from (36), and denoting the difference as $E_i := \mathbf{x}_{\epsilon,i} - \hat{\mathbf{x}}_{\epsilon,i}$, we obtain

$$E_{i+1} = E_i + \Delta t \left( \mathbf{f}_\epsilon(\mathbf{x}_{\epsilon,i}) - \mathbf{f}_\epsilon(\hat{\mathbf{x}}_{\epsilon,i}) \right) + (\Delta t)^2 \left( \hat{\mathbf{N}}_{\epsilon,i} \left( \hat{g}(\hat{\mathbf{T}}_{\epsilon,i}^\top \mathbf{f}_\epsilon(\mathbf{x}_{\epsilon,i})) - \hat{g}(\hat{\mathbf{T}}_{\epsilon,i}^\top \mathbf{f}_\epsilon(\hat{\mathbf{x}}_{\epsilon,i})) \right) \right) + \tau_i \Delta t. \tag{37}$$

and using the fact that $\mathbf{f}_\epsilon$ and $\hat{g}$ are Lipschitz continuous and denoting $L$ as the largest Lipschitz constant among these two functions, we have

$$\|E_{i+1}\|_\infty \leq (1 + L\Delta t + L^2(\Delta t)^2)\|E_i\|_\infty + \|\tau_i\|_\infty \Delta t,$$

where $\tilde{C} = \hat{C}|g|_{C^{\ell+1}(M^*)}$. Solving this discrete equation, we have,

$$\|E_i\|_\infty \leq (1 + L\Delta t + L^2(\Delta t)^2)^i \|E_0\|_\infty + \Delta t \sum_{j=1}^i (1 + L\Delta t + L^2(\Delta t)^2)^{i-j} \|\tau_{j-1}\|_\infty.$$

For time interval $[0, T]$ where $n\Delta t = T$, we have,

$$(1 + L\Delta t + L^2(\Delta t)^2)^{i-j} \leq e^{L(i-j)\Delta t} \leq e^{Ln\Delta t} = e^{LT}.$$

With this identity, it is clear that, with probability higher than $1 - \frac{1}{N}$,

$$\|E_i\| \leq e^{LT} \left( \|E_0\| + \tilde{C} T \Delta t \left( \frac{\log(N)}{N} \right)^{\frac{\min\{\ell+1, s\}}{d}} \right)$$

Since no error is committed at the initial condition, $E_0 = 0$ and we conclude that,

$$\|E_i\|_\infty \leq \tilde{C} T e^{LT} \Delta t \left[ \left( \frac{\log(N)}{N} \right)^{\frac{\min\{\ell+1, s\}}{d}} \right],$$

as $\Delta t \to 0$ and $N \to \infty$. Taking the expectation of square under $\mu$ and using the upper bound of the density in the Assumption D.2, the proof is complete. $\qquad\square$

With these results, the proof of Theorem 3.1 is rather straightforward, owing to the fact that,

$$\sqrt{\mathbb{E}_\mu \left[ \|\mathbf{x}_n - \hat{\mathbf{x}}_{\epsilon,n}\|^2 \right]} \quad \leq \quad \sqrt{\mathbb{E}_\mu \left[ \|\mathbf{x}_n - \mathbf{x}_{\epsilon,n}\|^2 \right] + \mathbb{E}_\mu \left[ \|\mathbf{x}_{\epsilon,n} - \hat{\mathbf{x}}_{\epsilon,n}\|^2 \right]}$$

$$\leq \quad \sqrt{\mathbb{E}_\mu \left[ \|\mathbf{x}_n - \mathbf{x}_{\epsilon,n}\|^2 \right]} + \sqrt{\mathbb{E}_\mu \left[ \|\mathbf{x}_{\epsilon,n} - \hat{\mathbf{x}}_{\epsilon,n}\|^2 \right]}, \tag{38}$$

and the upper bounds in Propositions D.1 and D.2 with $\ell = s$. When using the result from Proposition D.1, we note the notations $\mathbf{x}_n := \mathbf{x}(t_n) = \mathbf{x}(T)$ and $\mathbf{x}_{\epsilon,n} := \mathbf{x}_\epsilon(t_n) = \mathbf{x}_\epsilon(T)$ where $t_n = n\Delta t = T$.

Table 2: Model parameters of GMKRR for the 1D cases

| CASE | SE Kernel Bandwidth, $\gamma$ | Order of GMLS for NC, $\ell$ | Tangent Space Estimation, $s$ |
|---|---|---|---|
| $D = 0$ | 1.0 | 6 | 3 |
| $D = 0.1$ | 1.0 | 7 | 5 |
| $D = 0.3$ | 1.0 | 6 | 5 |
| $D = 0.5$ | 1.0 | 5 | 4 |

## E  ADDITIONAL EXPERIMENTAL DETAILS

All the reported results, including computational costs, are generated on a MacBook Pro Intel Core i7. All the training and prediction are run on CPU.

### E.1  A 1D DYNAMICS EXAMPLE

**Dataset.**  The solution to the scalar ODE,

$$\dot{\theta} = \frac{3}{2} - \cos(\theta),\ \theta(0) = 0$$

has a closed-form solution

$$\theta(t) = 2\tan^{-1}\left(\tan(\sqrt{5}t/4)/\sqrt{5}\right)$$

and using the given embedding

$$(x_1, x_2) = (r(\theta)\cos(\theta), r(\theta)\sin(\theta))$$

one can evaluate the exact trajectory $(x_1, x_2)$ at any $t > 0$, that lies on the manifold.

For a given $(K, D)$ combination, the trajectory is sampled over $[0, 128]$ with a step size of $\Delta t = 0.04$, which ends up with 3201 steps. The first 200 steps, which covers approximately 1.5 periods, are used for training; the test starts from $t = 0$ and covers all the 3201 steps.

**Hyperparameters of the models.**  In Fig. 1, the GMKRR model uses the SE kernel. The hyperparameters include the kernel bandwidth $\gamma$, order of GMLS for normal correction $\ell$, and order of GMLS for estimation of tangent space $s$; $\ell$ is only used for Euler+NC. A validation procedure is used to identify the best hyperparameters for the GMKRR model combined with a particular ODE solver as follows. A grid search is performed over $(\gamma, \ell, s) \in \{0.1, 0.5, 1.0, 5.0, 10.0\} \times \{3, 4, 5, 6, 7\} \times \{3, 4, 5, 6, 7\}$, and for each combination $(\gamma, \ell, s)$, the dynamics is solved using the chosen solver for 50 steps starting from $t = 0$ and the RMSE is recorded. When an initial value of $\gamma$ were selected, more rounds of refined grid search over $(\gamma, \ell, s)$ would be performed until no further improvement in prediction is observed. For example, if $\gamma = 1.0$ is selected from $\{0.1, 0.5, 1.0, 5.0, 10.0\}$, then a narrower range $(\gamma, \ell, s) \in \{0.8, 0.9, 1.0, 1.1, 1.2\} \times \{3, 4, 5, 6, 7\} \times \{3, 4, 5, 6, 7\}$ would be considered next. The hyperparameters associated with the lowest RMSE are used for the GMKRR model. In practice, GMKRR model prediction is found to be insensitive to the hyperparameters when Euler+NC is used. Hence, for a $(K, D)$ combination, the GMKRR model is trained using the optimal $\gamma$ and $s$ for RK2 and the optimal $\ell$ for Euler+NC. The detailed choices are provided in Table 2. This choice of hyperparameters gives a slight disadvantage on Euler+NC. However, as one has seen in Fig. 1b and Fig. 5 for a more complete prediction horizon, RK2 still fails miserably for longer prediction time in all cases, while Euler+NC consistently predicts the periods and amplitudes even though it is only first-order time accurate.

**Comparison with other baselines.**  In Fig. 1b (and Fig. 5 for the entire prediction time and other values of $D$), the NODE method (Chen et al., 2018) and the LDNet method (Regazzoni et al., 2024) are included as the baselines. The codes for these methods are publicly available

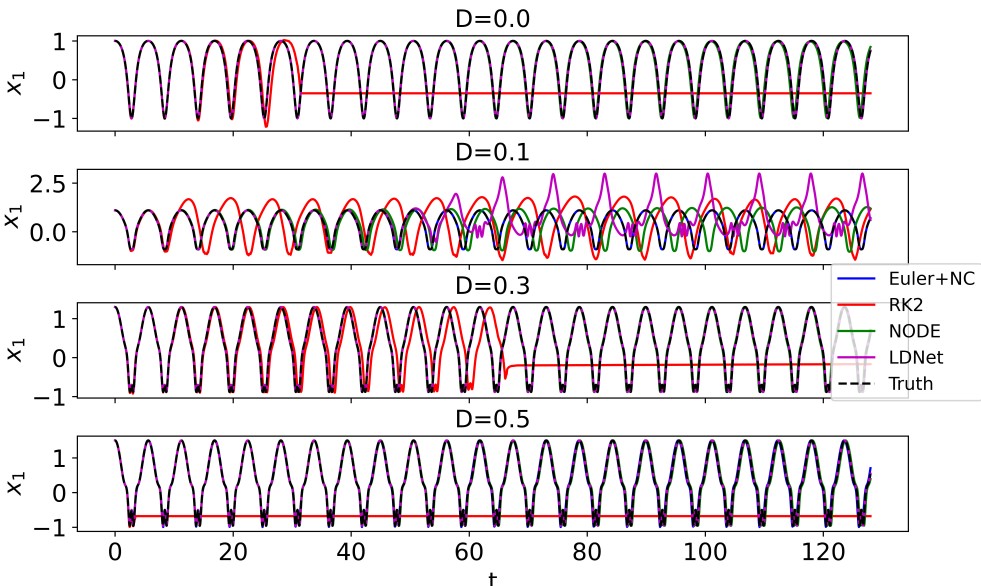

Figure 5: Predictions time for various values of $D$ (A more complete depiction of Figure 1b).

online (NODE[1] and LDNet[2]). For NODE, the network architecture consists of two inputs and two outputs, with two hidden layers of 32 neurons each and PReLU activation function. The NODE is time-integrated by RK4. The model is trained using Adam optimizer with a learning rate of $1 \times 10^{-4}$ over 1000 iterations, using a batch size of 20 trajectory segments; in each batch, the segments are randomly selected from the same training dataset used by GMKRR (one trajectory of 200 steps). For LDNet, the network architecture consists of a dynamics network and a reconstruction network. The former has two inputs (number of states, i.e., ambient dimensions) and two outputs (latent variables), with a hidden layer of 20 neurons and tanh activation function; the latter has three inputs (latent variables and index of ambient dimensions) and one output (the value of one state), with a hidden layer of 20 neurons and tanh activation function. The LDNet is designed to be time-integrated by the forward Euler method. The model is trained using first Adam optimizer with a learning rate of $1 \times 10^{-2}$ over 200 iterations, and then BFGS optimizer for 1300-4000 iterations, until the loss drops to below $5 \times 10^{-9}$. However, it is worth noting that we failed to train LDNet using the same size of training dataset used by GMKRR, and the reported prediction results are based on the training of twice the size of the dataset used in training GMKRR.

Besides the predictions of the last 6 periods in Fig. 1 and the entire time in Fig. 5, the test RMSEs are provided in Table 3. While the RMSEs of Euler+NC and NODE are comparable for the most of $D$'s, the performance of the Euler+NC method is clearly better than the NODE when $D = 0.1$. For this latter case, NODE fails to maintain the geometrical constraint over long time prediction. The LDNet behaves in an irregular manner too: it achieves the best accuracy for $D = 0, 0.3, 0.5$, but fails to predict the true trajectory after about 50 model time unit for $D = 0.1$. Furthermore, we note that when only one latent variable is used (i.e., the same as the intrinsic dimension) LDNet could not produce any useful predictions. Furthermore, the average training and prediction costs for all the methods are given in Table 4. While the prediction costs of the methods are comparable, the training costs of NODE and LDNet are 3-4 orders of magnitude higher than the GMKRR methods.

**Effect of kernels.** Besides the results in Fig. 1, the effect of kernels on the GMKRR model is also examined. We consider the most challenging case of $D = 0.5$, and four kernels:

1. SE kernel: $\rho(\mathbf{x}, \mathbf{x}') = \exp(-\left\| \mathbf{x} - \mathbf{x}' \right\|^2 / \gamma)$.
2. Matérn 5/2 (M52): $\rho(\mathbf{x}, \mathbf{x}') = (1 + \sqrt{5}d + 5/3d^2) \exp(-\sqrt{5}d)$.

---

[1] https://github.com/rtqichen/torchdiffeq
[2] https://github.com/FrancescoRegazzoni/LDNets

Table 3: RMSE values for different cases

| Case (D) | Euler+NC | RK2 | NeuralODE | LDNet |
|---|---|---|---|---|
| Case 1 (D = 0.0) | 0.0184 | 1.5178 | 0.4372 | 0.0001 |
| Case 2 (D = 0.1) | 0.0333 | 1.9530 | 1.5649 | 2.8000 |
| Case 3 (D = 0.3) | 0.0224 | 1.6325 | 0.2349 | 0.0002 |
| Case 4 (D = 0.5) | 0.3842 | 2.0433 | 0.4450 | 0.0157 |

Table 4: Average training and prediction times for the 1D example.

| Method | Training Time (s) | Prediction Time (s) |
|---|---|---|
| Euler+NC | 0.1077 | 5.159 |
| RK2 | | 3.363 |
| Neural ODE | 580.9 | 1.003 |
| LDNet | 84.81 | 2.574 |

3. Matérn 3/2 (M32): $\rho(\mathbf{x}, \mathbf{x}') = (1 + \sqrt{3}d) \exp(-\sqrt{3}d)$.

4. Matérn 1/2 (M12): $\rho(\mathbf{x}, \mathbf{x}') = \exp(-d)$

where $d = \|\mathbf{x} - \mathbf{x}'\|/\gamma$.

After a cross-validation similar to the one outlined at the beginning of this section, the hyper-parameters for different kernels are listed in Table 5. Unlike the case in Fig. 1, the bandwidth parameter $\gamma$ for each kernel is also obtained by cross-validation, to ensure fair comparison among the kernels. The prediction is shown in Fig. 6, and the RMSE's are listed in Table 5. Clearly, there are no significant differences in terms of prediction skills among the different kernels; this confirms that the framework is not restrictive to only the SE kernel. This is as expected since this problem has smooth vector fields. However, we do note that it is very likely that there are some examples of certain regularity where it is advantageous to employ the Matérn kernels.

Table 5: Hyperparameters and RMSE's for models with different kernels.

| Kernel | SE Kernel Band-width, $\gamma$ | Order of GMLS for NC, $\ell$ | Tangent Space Es-timation, $s$ | RMSE |
|---|---|---|---|---|
| SE | 1.90 | 5 | 4 | 0.4068 |
| M52 | 0.28 | 3 | 4 | 0.5861 |
| M32 | 0.56 | 3 | 4 | 0.7860 |
| M12 | 0.65 | 3 | 5 | 0.5559 |

**Effect of accuracy of tangent space.** Among the examples considered in this study, the 1D example is the only one where all the information is known exactly; this includes the manifold parameterization and tangent space as well as the dynamics. Hence, this example serves as a good case for the numerical verification of the convergence bound for the Euler+NC method in Theorem 3.1, which is repeated below,

$$\sqrt{\mathbb{E}_\mu \left[ \|\mathbf{x}_n - \hat{\mathbf{x}}_{\epsilon,n}\|^2 \right]} \le C \left( \epsilon + \Delta t \left( \frac{\log(N)}{N} \right)^{\frac{\ell}{d}} \right),$$

where $C \propto e^{LT}$.

The verification is designed as follows. First, to avoid the effect of $\epsilon$, i.e., the error in estimated dynamics, the true dynamics in ambient space for the 1D example is used. In this case, the error bound should be $O((\log(N)/N)^\ell)$ where the constant depends on $\Delta t$. Next, we choose a small time step size $\Delta t = 10^{-6}$ and a short prediction horizon $T = 10^{-4}$; this is to avoid the dominance of errors due to time discretization and constant $C$ in the bound. Subsequently, a series of datasets

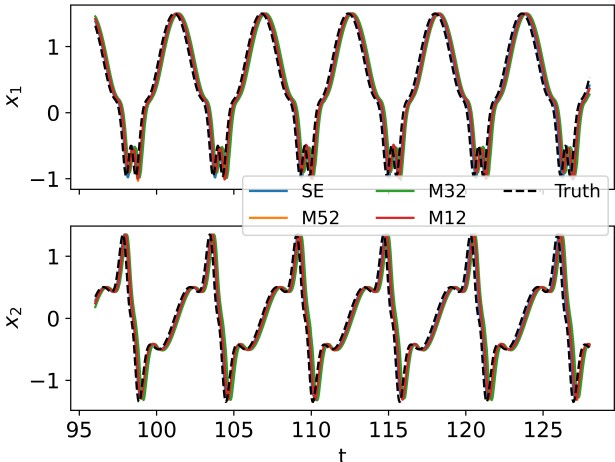

Figure 6: Model predictions using different kernels.

of size $N_i = 160 \times 2^i$, $i = 1, 2, 3, 4, 5$ are generated. For each $i$, $N_i$ data points are sampled from the manifold, and the dataset is used for the tangent space estimation and the normal correction in the Euler+NC method. Lastly, 100 test trajectories are considered, whose initial conditions are randomly sampled on the manifold. The RMSE of prediction errors at the last step is recorded for each $N_i$. The convergence plot is shown in Fig. 7, which shows the cases of $\ell = 2, 3$. The two cases closely follow the theoretical convergence rates.

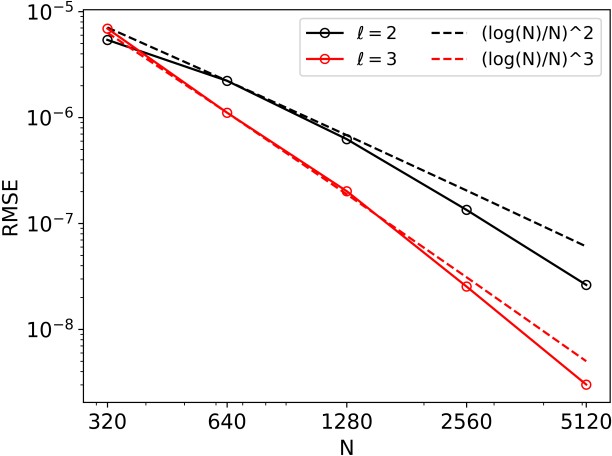

Figure 7: Convergence of Euler+NC method given true dynamics.

### E.2 A 2D CAVITY FLOW PROBLEM

**Dataset.** The 2D cavity flow dataset is publicly available from PySINDy (2024) and is generated from one Direct Numerical Simulation (DNS) solution of a 2D laminar cavity flow problem. According to Callaham et al. (2022), the original DNS solution has 30000 steps, with a step size of $\Delta t = 0.01$s. The first 4000 steps are used to perform a linear dimension reduction by Proper Orthogonal Decomposition (POD). The leading 64 POD modes are retained and all the 30000 steps of solutions are projected onto the 64 POD coordinates. Hence, the available dataset consists of one single trajectory of 30000 steps, and the ambient dimension is $n = 64$.

For fairness in comparison, the SINDy, M-SINDy, and GMKRR models are trained using the first 4000 data points. For testing, the remaining 26000 data points are split into 13 trajectories of 2000 steps.

**Hyperparameters of the models.** First, the setup of the GMKRR algorithm is discussed. The intrinsic dimension of the dataset is estimated to be 2.38 using a kernel-based algorithm (see Appendix B); this indicates that the dataset approximately lives on a two-dimensional manifold, and hence $d = 2$ is chosen. The rest of the hyperparameters in GMKRR are chosen via a validation procedure similar to that of the 1D example. The SE kernel bandwidth $\gamma = 10$, polynomial order of GMLS for NC $\ell = 3$, and the tangent space estimation is done by local SVD and denoted $s = 0$.

The SINDy algorithm is used as a baseline case. Due to the limitation on the curse of dimensionality, the SINDy model is only trained to predict the first 6 POD coordinates, using third-order polynomial features.

The M-SINDy algorithm is an existing approach proposed in Callaham et al. (2022) to learn dynamics on manifold. This algorithm first applied Dynamic Mode Decomposition (DMD) to represent the original 64-dimensional dynamics by temporally-coherent modes $\mathbf{U}$ and DMD coordinates $\mathbf{z}(t)$, $\mathbf{x}(t) = \mathbf{Uz}(t)$. Then 4 DMD coordinates, denoted $\hat{\mathbf{z}}$, are manually selected by physical intuition to represent the manifold on which the data points reside. Subsequently, standard SINDy with third-order polynomial feature is applied to learn the dynamics $\dot{\hat{\mathbf{z}}} = \mathbf{f}(\hat{\mathbf{z}})$, while another fifth-order polynomial model $\mathbf{z} = \mathbf{g}(\hat{\mathbf{z}})$ is trained to estimate the full DMD coordinates using $\hat{\mathbf{z}}$. In the prediction stage, the ODE $\dot{\hat{\mathbf{z}}} = \mathbf{f}(\hat{\mathbf{z}})$ is solved to generate the dynamics in $\hat{\mathbf{z}}$, then the POD coordinates are recovered by DMD modes $\mathbf{x} = \mathbf{Ug}(\hat{\mathbf{z}})$. In this paper, the open-source implementation PySINDy (2024) of M-SINDy for the cavity flow problem is used to generate the results for comparison.

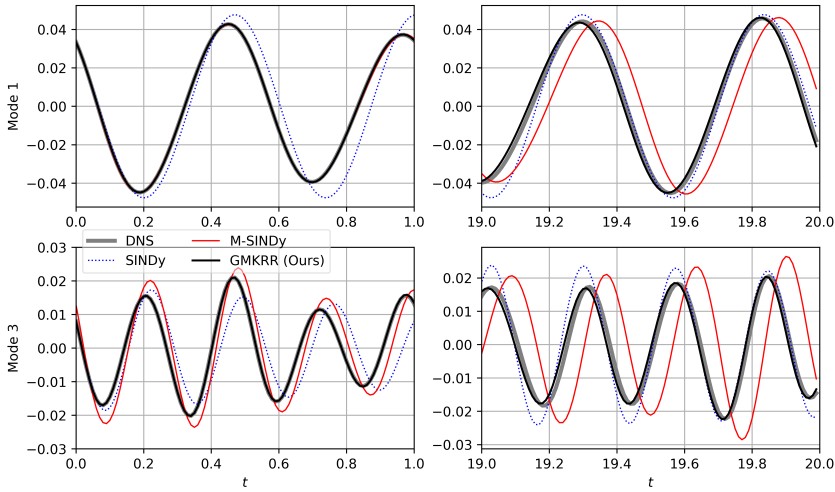

Figure 8: Comparison of predictions for a representative trajectory.

**Additional results.** The predicted dynamics from SINDy, M-SINDy, and GMKRR are compared for one typical trajectory in Fig. 8, which shows the 1st and 3rd POD coordinates at the first second and the last second of a test trajectory. The Mode 1 results show typical characteristics of the three methods. The SINDy starts to deviate from the truth from the beginning, as it does not satisfy the manifold constraints. Both M-SINDy and GMKRR predict the trajectory of the solutions accurately in the beginning, but over time a discernible phase shift is observed in the M-SINDy predictions. The Mode 3 results show that, using the same amount of training data (4000 points), GMKRR captures the oscillation amplitudes much better than M-SINDy.

The convergence study of GMKRR is performed on the number of samples, $N$. The correct functioning of GMKRR algorithm requires the training dataset to capture the entire manifold, yet for the quasi-steady cavity dynamics, the complete coverage of the manifold requires over 3000

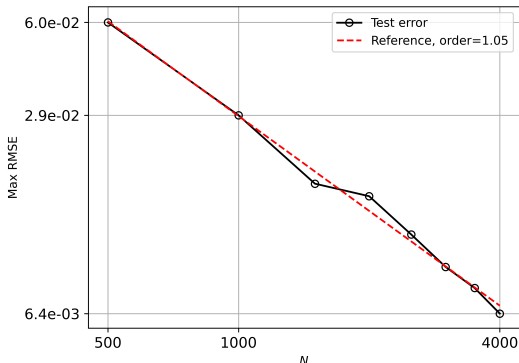

Figure 9: Convergence study of GMKRR: Max RMSE as a function of the number of training samples $N$.

steps. Hence in the convergence study the following strategy is employed. The original training dataset of 4000 data points are split into 100 shorter trajectories of 40 steps. A dataset of $N = 100K$ data points is formed by choosing the first $K$ steps of each shorter trajectory. For convergence study, a series of datasets are generated with $K = 5, 10, 15, 20, 25, 30, 35, 40$, and for each model the average error of the test trajectories is recorded. The results are shown in Fig. 9, indicating RMSE $\sim O(N^{-1})$; this convergence rate aligns with standard kernel regression methods.

### E.3    KURAMOTO-SIVASHINSKY DYNAMICS

**Datasets.**    The KS dynamics is governed by the following partial differential equation (PDE),

$$u_t + u u_x + u_{xx} + \nu u_{xxxx} = 0, \quad x \in [0, L], \ t \geq 0.$$

Here we consider a periodic boundary condition $u(t, 0) = u(t, L)$ and a tunable parameter $\nu$. When $L = 2\pi$ and given an appropriate initial condition $u(0, x)$, on a discretized grid of 64 points, one obtains beating dynamics for $\nu = \frac{16}{337}$ and travelling dynamics for $\nu = \frac{4}{87}$ (Rowley & Marsden, 2000).

In Floryan & Graham (2022), both the beating and travelling dynamics are considered; the models are trained using short time intervals of training data (e.g., 1-2 periods of oscillation) and predicted the dynamics over long time intervals (e.g., over 10 periods of oscillation). However, in their open-source repository, only the training datasets are provided. To quantitatively assess the long-term prediction errors, we have to regenerate the complete long-term datasets. Furthermore, since the numerical methods employed by Floryan & Graham (2022) and us differ, we take the following procedure to ensure the consistency of datasets.

Our numerical solution of the KS equation is obtained using the standard Exponential Time Differencing method. Next,

1. Beating dynamics: We choose a step size of 0.0001, and the initial condition to be the last step in CANDyMan's training dataset of beating dynamics. The simulation is run for 75000 steps, and the initial 25000 steps of transients are discarded. The remaining 50000 steps are subsampled with a step size of 0.01, so that eventually 500 steps are saved as the dataset. The period of beating dynamics is approximately 45 steps.

2. Travelling dynamics: We choose a step size of 0.001, and take the same strategy as above for the initial condition. The simulation is run for 180000 steps, and the initial 40000 steps of transients are discarded. With a sampling step size of 0.01, 14000 steps are saved as the dataset. The long period of travelling dynamics is approximately 5000 steps.

Both the GMKRR and CANDyMan algorithms are tested on these newly-generated datasets. For CANDyMan, since the physical behaviors of the new datasets are the same as the original ones, we employed the identical model architectures and all hyperparameters (e.g., number of neurons and training epochs) from the original paper. The hyperparameters of GMKRR models are found through a cross-validation procedure and listed in Table 6. Note that here the initial guess of the bandwidth parameter is determined using $\gamma^*$ defined in App. B.2.

**Beating dynamics.** The GMKRR and CANDyMan models are trained using the first 100 steps of dataset, which correspond to approximately 2 periods of oscillations. The prediction is computed by taking the first step of the dataset as the initial condition and performing time stepping for 500 steps, i.e., approximately 11 periods.

The predicted spatiotemporal dynamics are shown in Fig. 10. While both models seem to produce visually similar results, as shown in Fig. 10a, GMKRR does show one order of magnitude lower error than CANDyMan. The sources of error in CANDyMan are explained via the comparison of the time histories of $u$ at two spatial locations; in Fig. 10b, the last two periods of time history are shown. Clearly, CANDyMan shows some phase error after 11 periods of prediction. Furthermore, in the second panel, while the amplitude of oscillation is low, CANDyMan shows non-physical highly-oscillatory response; this is attributed to the errors in the autoencoder of CANDyMan.

**Travelling dynamics.** As explained in Sec. 5.2, five models are considered for this case: CANDy-Man, NODE, LDNet, GMKRR-Full, and GMKRR-FFT. The GMKRR-Full model is trained using every other step in the first 6000 steps of data (so a total of 3000 steps) without assuming any structure in the model. The subsampling of every other step is to reduce the computational cost of the kernel regression. The CANDyMan, NODE, LDNet, and GMKRR-FFT employ the following special data preprocessing step. Denote the data sample at time $t$ as $\mathbf{u}(t) \in \mathbb{R}^{64}$. The preprocessing starts with a discrete Fourier transform to decompose the data $\mathbf{u}(t) = \text{Re}(\hat{\mathbf{u}}(t) \exp(i\phi(t)))$ into a standing wave $\hat{\mathbf{u}}(t)$ and its phase shift $\phi(t)$. The standing wave and phase dynamics capture the fast and slow timescales, respectively, and both have an intrinsic dimension of 1. After the preprocessing step, the four models only needed the first 100 steps to train for the standing wave dynamics $\dot{\hat{\mathbf{u}}} = f_{NN}(\hat{\mathbf{u}})$ as well as the change in phase per time step $\Delta\phi = g_{NN}(\hat{\mathbf{u}})$.

The hyperparameters for CANDyMan are the same as the original paper, where its atlas-of-charts approach is applied to $f_{NN}$ and $g_{NN}$ is a vanilla fully-connected NN (for each chart). For the NODE, $f_{NN}$ consists of 64 inputs and 64 outputs, with two hidden layers of 128 neurons each and tanh activation function, and the rest is the same as the 1D case, except that 2000 iterations were used in training; $g_{NN}$ consists of one hidden layer of 32 neurons and tanh activation function, and is trained by Adam optimizer with learning rate $10^{-3}$. The LDNet is the same as NODE, except the $f_{NN}$ is replaced by the latent dynamics. The dynamics network has 64 inputs and two outputs (latent variables), with two hidden layers of 8 neurons each and tanh activation function; the reconstruction network three inputs (latent variables and $x$-coordinate) and one output (the value of one state), with two hidden layers of 8 neurons each and tanh activation function. The rest is the same as the 1D case, except that 6000 iterations were used in training. In this case LDNet use the same dataset as other models. Note that LDNet would fail to produce reasonable predictions if only one latent variable (i.e., the intrinsic dimension) is used.

Figure 11a shows the initial short period of dynamics. All models perform reasonably well, and all captures the short time-scale with low errors. Yet, the GMKRR models either outperform or are on par with other baselines in terms of prediction accuracy. Figure 11b shows the long-term dynamics. There is a clear difference in the error level across the three models, that is attributed to the mismatch in the long time-scale. GMKRR-FFT almost captures exactly the long time-scale; this results in the lowest error, that is almost one order of magnitude lower than all other baselines, even though these models leverage the same prior knowledge. The comparison highlights that, while the models that do not account for geometrical constraints (i.e., NODE and LDNet) can capture short-term dynamics well, their accuracy quickly deteriorate over time. The GMKRR outperforms other models, which is attributed to its capability to model the dynamics using exactly the intrinsic dimension.

### E.4 REACTION-DIFFUSION DYNAMICS

The reaction-diffusion dynamics is governed by the following coupled 2D PDEs,

$$u_t = [1 - (u^2 + v^2)]u + \beta(u^2 + v^2)v + d_1(u_{xx} + u_{yy})$$
$$v_t = -\beta(u^2 + v^2)u + [1 - (u^2 + v^2)]v + d_2(v_{xx} + v_{yy})$$

on a square domain $(x, y) \in [-10, 10] \times [-10, 10]$ with homogeneous Neumann boundary conditions and parameters $d_1 = d_2 = 0.1$, $\beta = 1$. We follow the same numerical solution method as in CANDyMan by directly transcribing the MATLAB-based solver in its public repository into Python

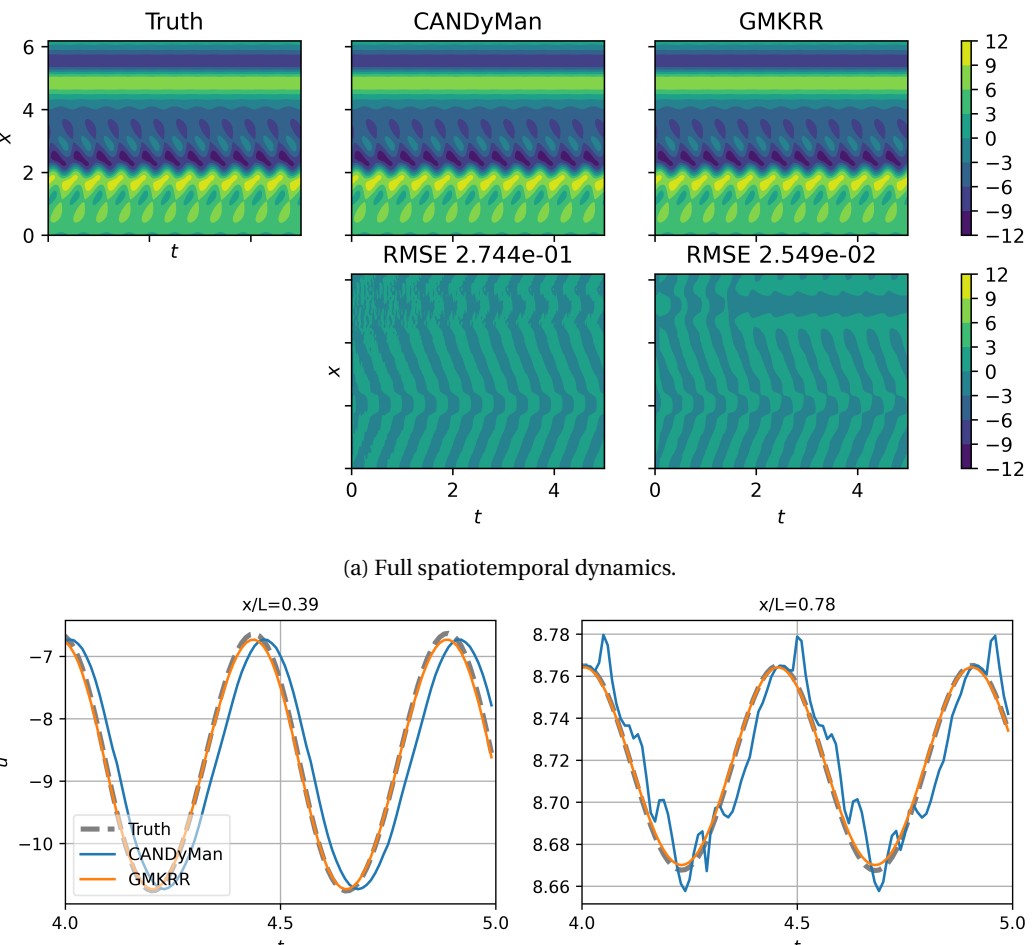

(a) Full spatiotemporal dynamics.

(b) Dynamics at select spatial coordinates.

Figure 10: The KS beating dynamics.

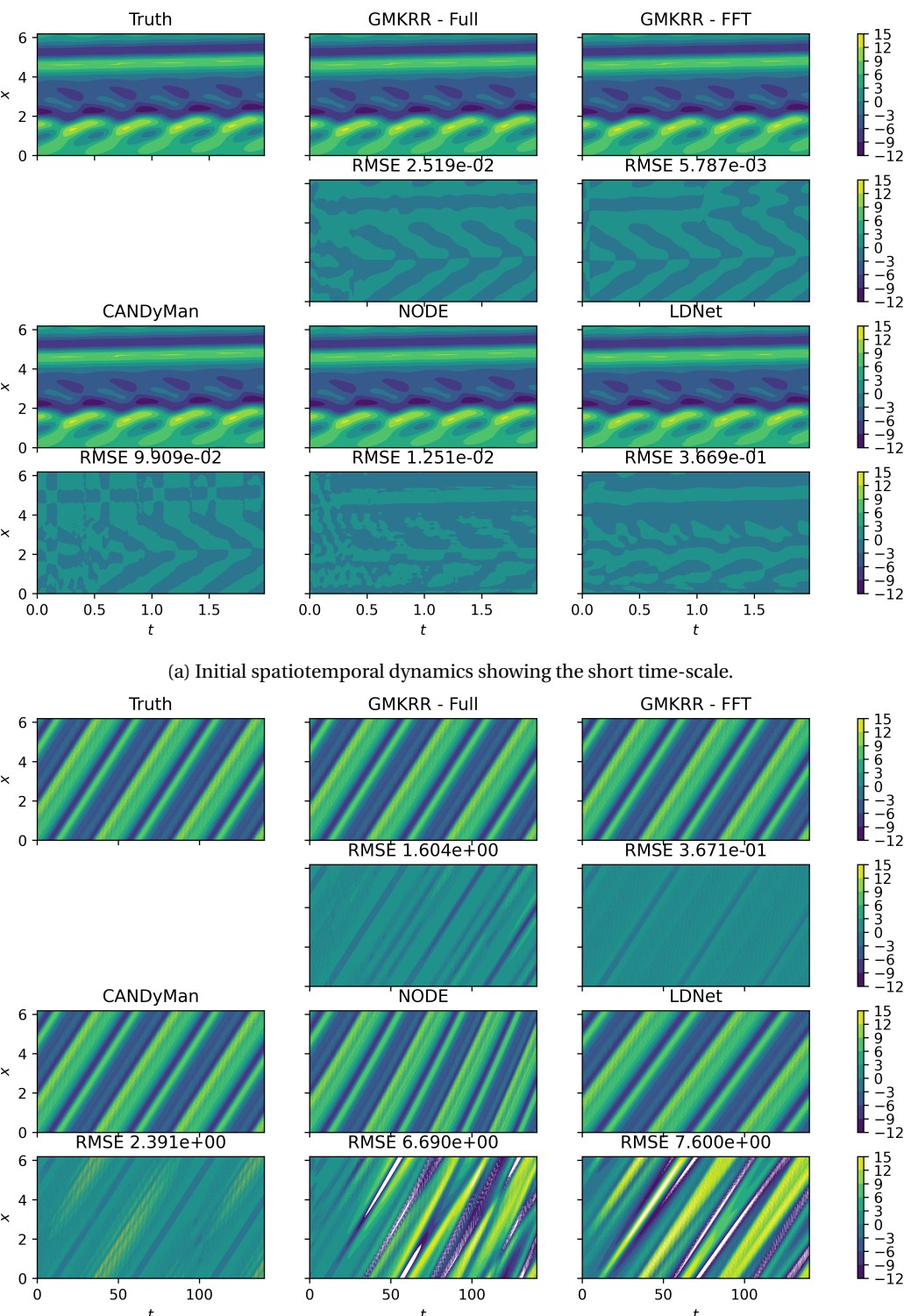

(a) Initial spatiotemporal dynamics showing the short time-scale.

(b) Full spatiotemporal dynamics showing the long time-scale.

Figure 11: The KS travelling dynamics.

Table 6: Model parameters of GMKRR for the CANDyMan cases

| CASE | MODEL | Intrinsic Dimension, $d$ | SE Kernel Bandwidth, $\gamma$ | Order of GMLS for NC, $\ell$ | Tangent Space Estimation, $s$ |
|---|---|---|---|---|---|
| KS Beating | GMKRR | 1 | 440.163 | 4 | 0 |
| KS Travelling | GMKRR-Full | 3 | 98.0816 | 3 | 0 |
| | GMKRR-FFT | 1 | 61.47 | 4 | 0 |
| Reaction-Diffusion | GMKRR-Full | 1 | 31000.6 | 4 | 0 |
| | GMKRR-PCA-T0 | 1 | 31319.5 | 4 | 0 |
| | GMKRR-PCA-T4 | 1 | 31319.5 | 4 | 4 |

using NumPy/SciPy. The domain is discretized into a uniform $101 \times 101$ grid, so the solution at one time step is of dimension 20402. A step size of 0.05 is used, and 10200 steps are simulated; the first 5200 steps are discarded to avoid transients. A period of response is approximately 72 steps, and hence the full dataset contains nearly 70 periods. All the models are trained using the first 200 steps and tested on the full 5000 steps of data.

Figures 12a and 12b show results at the last time step. While visually all spirals seem similar, CANDyMan does have errors that are two orders of magnitude larger than GMKRR methods. In addition, the high-order tangent space estimation shows further and dramatic improvement over the other GMKRR models. The large error in CANDyMan is explained by Fig. 12c. The CANDyMan model does not accurately capture the period, and hence results in accumulation of error over time.

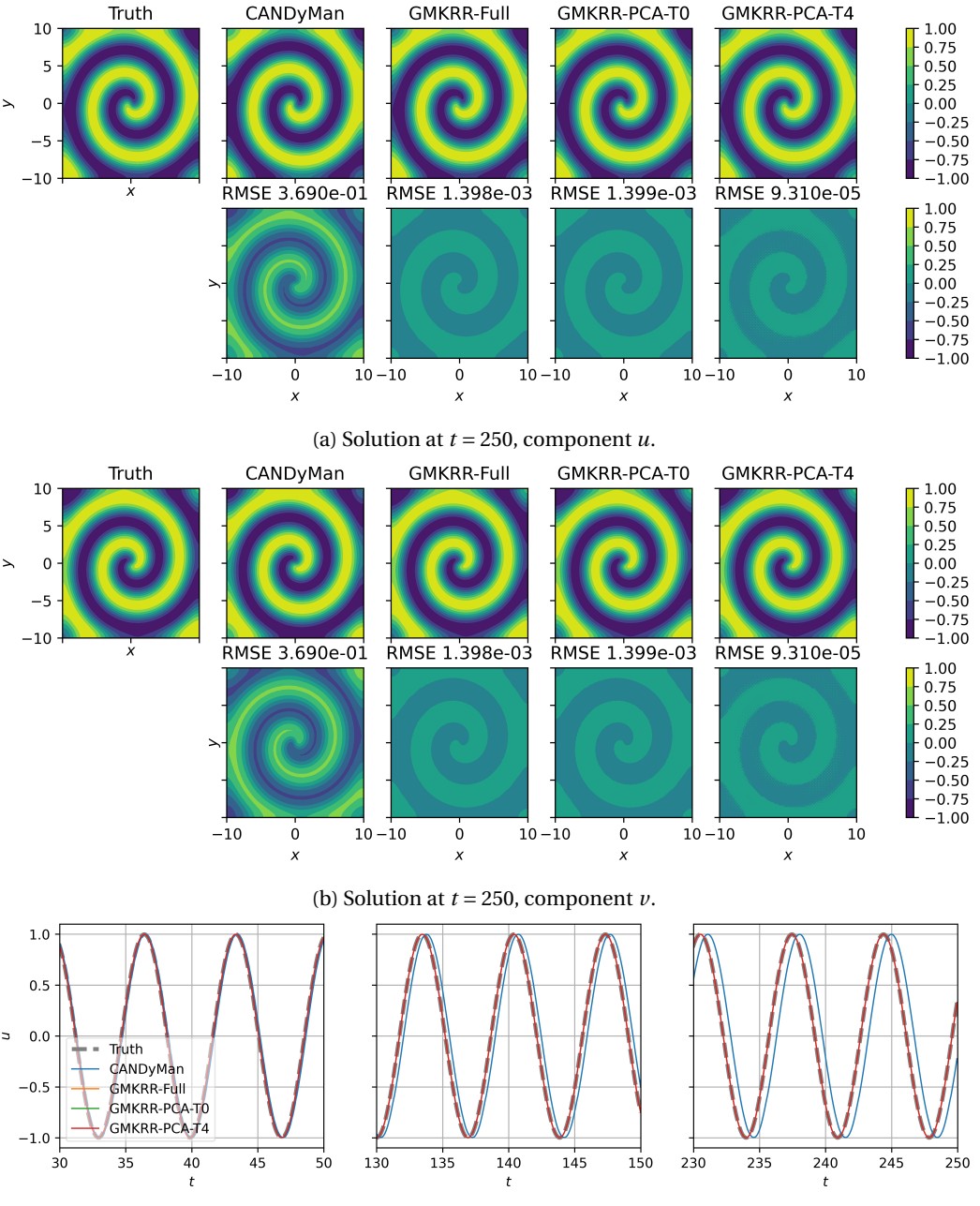

(a) Solution at $t = 250$, component $u$.

(b) Solution at $t = 250$, component $v$.

(c) The $u$ dynamics at $(x, y) = (10, -10)$; all GMKRR models overlap.

Figure 12: Reaction-diffusion dynamics.

