# OpenReview forum: "Learning vector fields of differential equations on manifolds with geometrically constrained operator-valued kernels"
_ICLR.cc/2025/Conference — ICLR 2025 Spotlight_

### Official Review · Reviewer_YaJd · 2024-10-28

**Soundness:** 4
**Presentation:** 3
**Contribution:** 4
**Rating:** 8
**Confidence:** 3

**Summary:**

The paper presents an approach for learning ODEs on manifolds using kernel methods such that learnt ODE solutions are constrained to lie on the learnt manifold. They also present a method for solving ODEs on manifolds more accurately than standard RK based approaches in terms of how closely solutions lie on said manifold.

**Strengths:**

- Learning ODEs constrained to evolve on manifolds using linear methods is a beautiful idea.
- The geometry-preserving time integrator is well-motivated and could have impacts well beyond learning differential equations from data.
- The numerical studies convincingly demonstrate the proposed approach is better than the SINDy-style approaches they compare to.

**Weaknesses:**

- While theoretically interesting, I think the authors could have done a better job of motivating their approach in the introduction from a practical perspective. In particular, it wasn't clear to me that the standard approach of learning latent differential equations, is limited in practice by the fact that learnt differential equations will deviate from the learnt manifold. Since the standard approach is is so widely used, providing concrete references / studies which demonstrate these issues in practice would have been helpful for motivating the work.

- I think it's arguable that the one of the biggest challenges with SINDy-style algorithms is that they struggle in the presence of noise, for example:

    - Messenger, Daniel A., and David M. Bortz. "Weak SINDy for partial differential equations." Journal of Computational Physics 443 (2021): 110525

    - Fasel, Urban, et al. "Ensemble-SINDy: Robust sparse model discovery in the low-data, high-noise limit, with active learning and control." Proceedings of the Royal Society A 478.2260 (2022): 20210904.

    Since your approach also requires estimating derivatives using finite differences, it would have been helpful to include some discussion on this issue.

- You mentioned that one of the goals of your work as compared to standard approaches for learning latent differential equations is to learn interpretable models. As I understand it, SINDy-style models are interpretable because they learn sparse approximations to differential equations. Since you relax the sparsity requirement, I struggled to understand how your proposed approach is more interpretable than neural-network based models.


*Additional feedback. To be clear, these points are here to help and will not be a part of the decision assessment.*

- When introducing SINDy, I think it would be helpful to include a reference to this work since I think they also proposed to learn sparse dynamics using dictionaries of basis functions in 2011.

    - Wang, W.X., et al. Predicting catastrophes in nonlinear dynamical systems by compressive sensing. Phys. Rev. Lett. 106, 154101 (2011).

- The results from your numerical studies would be made stronger by including comparison to learning latent ODEs with neural networks, for example:

    - Toth, P., Rezende, D. J., Jaegle, A., Racanière, S., Botev, A., & Higgins, I. (2019). "Hamiltonian generative networks." arXiv preprint arXiv:1909.13789.
    - Chen, Ricky TQ, Yulia Rubanova, and David Duvenaud. "Latent ODEs for irregularly-sampled time series." Advances in Neural Information Processing Systems 32 (2019): 3.

**Questions:**

1. How do you view your approach as being more interpretable than neural-network based methods?
2. How do you suspect you suspect your approach will scale for problems with higher dimensional submanifolds?
3. How do suspect your method will perform in the presence of significant observation noise given that you are required to estimate the manifold and the time derivatives on the manifold from data?

---

> ### Author Response · Authors · 2024-11-25
> **Formal response**
>
> Thank you for taking time to read the paper and your helpful comments.
>
> 1. Weakness, "While theoretically interesting ..."
>
> **Response**: Thank you for raising this very important point. In responding to your concern 5 below, we added LDNet by Regazzoni et al. 2024 as a baseline method and we verified this claim. Specifically, we found that if we specify LDnet with a latent dimension equal to exactly the intrinsic dimension, LDnet does not work. Second, in some cases, as shown in Figure 1(b) or Figure 5 for $D=0.1$, we note that the LDNet solutions can deviate from the manifold after a long time integration. We also verified the same conclusion in the KS travelling waves example (see Figures 4(j), 4(k) and 11). Based on these results, we adjusted the last two sentences in the 2nd paragraph of the introduction to convey this point.
>
> 2. Weakness, "... presence of noise ..."
>
> **Response**: You completely see our laundry list. This is indeed our next project. We consider replacing the vector field estimation with a loss function induced by the weak-SINDy type approach mentioned in these papers. We foresee replacing the standard local SVD with a robust PCA by Candes et al. 2011. The local GMLS projection in the prediction step needs to be adapted to noisy data as well, and we plan to replace the GMLS regression with a Gaussian process regression, which allows for uncertainty quantification.
>
> Due to space limitations, we will briefly mention the challenge with noisy data and mention possible remedies in the conclusion.
>
> 3. Weakness, "interpretability"
>
> **Response**: We agreed with you that the kernel approach is also not interpretable; we removed this sentence from the abstract and introduction.
>
> 4. Weakness-additional, "When introducing SINDy ..."
>
> **Response**: Thank you for the suggestion, we added the suggested references.
>
> 5. Weakness-additional, "The results from your numerical studies would be made stronger ..."
>
> **Response**: We tried to test the method from the second paper you mentioned above without any success. Unfortunately, we found that it is nontrivial to adjust the code to our setup. To compensate, we added two baseline methods in the revised manuscript: we provide two additional baseline methods: The Neural ODE (NODE) approach from Chen et al. 2018 and the Latent Dynamics Network (LDNet) from Regazzoni et al. 2024. See the updated Figure 1, where we reported the prediction for longer times (doubling the time in the previous report). The prediction for the entire time interval is shown in Fig 5. While the main discussion with added results can be found in Section 3.1 and Appendix E.1, let us highlight several new findings:
>
> + Euler+NC prediction is more accurate than NODE in all cases (see Table 4 for RMSE comparison). NODE fails to follow the manifold in a long-term prediction for the case $D=0.1$.
> + LDNet also fails to follow the manifold in a long-term prediction for $D=0.1$, but very accurate for $D=0.5$.
> + As we mentioned in the updated introduction (second paragraph), LDNet requires a latent variable of dimension larger than the intrinsic 1-dimension in this problem and sometimes produces solutions that do not lie on the manifold (as in the case of $D=0.1$).
> + For this example, we found that LDNet does not work if we use the same data set used for other methods (the reported results are obtained using twice the size of training data of the other methods).
>
> We also added these two baselines on the travelling waves example. Table 1 and Figure 4(j), (k) are updated, the paragraph discussing KS travelling wave in p.9 is also updated, and more detailed results and experiment configurations are presented in Appendix E.4 (see Fig 11). We found that both NODE and LDNet not only have high training costs, their predictions are less accurate compared to the proposed GMKRR.
>
> 6. Questions:
>
> 6(a). "interpretability"
>
> **Response**: This is addressed in bullet point 3 above.
>
> 6(b). "... scale for problems with higher dimensional submanifolds ..."
>
> **Response**: We have added a new section discussing the complexity of the algorithm in Appendix C. We have also added a new Appendix A.4 that discusses an algorithm to circumvent the computational bottleneck in realizing the normal vectors for very high ambient dimension as shown in one of our examples, $n=O(10^5)$.
>
> 6(c). "... noise ..."
>
> **Response**: This is addressed in bullet point 2 above.

---

> > ### Author Response · Authors · 2024-11-25
> > **References**
> >
> > + Emmanuel J Candès, Xiaodong Li, Yi Ma, and John Wright. Robust principal component analysis. Journal of the ACM (JACM), 58(3):1–37, 2011.
> > + Ricky TQ Chen, Yulia Rubanova, Jesse Bettencourt, and David K Duvenaud. Neural ordinary differential equations. Advances in neural information processing systems, 31, 2018.
> > + Francesco Regazzoni, Stefano Pagani, Matteo Salvador, Luca Dede, and Alfio Quarteroni. Learning the intrinsic dynamics of spatio-temporal processes through latent dynamics networks. Nature Communications, 15(1):1834, 2024.

---

> > ### Comment · Reviewer_YaJd · 2024-11-26
> >
> > Thank you for your thoughtful response and I look forward to reading your follow up work! I'll keep my score the same. I think this is a good paper which would be very interesting to ICLR 2025 readers.

---

> > > ### Author Response · Authors · 2024-11-26
> > >
> > > Thank you for your support and insightful comments that help us improved the manuscript.

---

### Official Review · Reviewer_zhfF · 2024-11-01

**Soundness:** 4
**Presentation:** 3
**Contribution:** 3
**Rating:** 8
**Confidence:** 3

**Summary:**

This paper overcomes some of the limitations of SiNDY and CandyMan to learn ODEs with solutions represented manifolds. The authors show that by enforcing geometric constraints, they can improve efficiency in training as well as accuracy compared to other methods on toy examples by ensuring solutions lie close to a lower dimensional submanifold that can express the data.

**Strengths:**

The paper is well written and provides detailed theoretical justification for their framework with clear numerical experiments highlighting the proposed method’s superiority to state-of-the-art works in this area. They benchmark with several new methods that are suitable for the problems they focus on and outperform in both training efficiency and accuracy.

**Weaknesses:**

The authors mention that their approach overcomes limitations of other algorithms in regard to making predictions on out of distribution data, yet they don’t detail the extent of which their method can practically handle problems where the manifold doesn’t exhibit properties or yield constraints that allow you to represent your solutions on tangent planes of low dimensional submanifolds. It would be beneficial to provide specific examples of manifold types or problem domains where their method might struggle.

**Questions:**

Are there specific types of out-of-distribution scenarios where the method is expected to perform well or poorly?

Can this method be used on domains or solutions that do not have or allow periodic behaviors that reduce the intrinsic dimensionality of the original solution manifold? It would be beneficial if the authors could provide a discussion on how the proposed method might be adapted for non-periodic systems.

---

> ### Author Response · Authors · 2024-11-14
> **question on the comment**
>
> Thank you for your support on this work. Before we respond to your question, we have a question. Can you specify which lines in the paper suggest that we made claims about out-of-distribution scenario?

---

> > ### Comment · Reviewer_zhfF · 2024-11-18
> >
> > No problem. Based on line 99, I was using out-of-sample and out-of-distribution as the same term. Sorry for the confusion.

---

> ### Author Response · Authors · 2024-11-25
> **Formal response**
>
> 1. Weakness, together with Question: "... out-of-distribution scenarios ..."
>
> **Response**: To clarify, we do not claim the method to work for out-of-distribution. As we understood, out-of-sampling corresponds to new sample points from the same distribution that is not used for training. To avoid this confusion, we rephrase this sentence.
>
> 2. Question: "... solutions that do not have or allow periodic behaviors ..."
>
> **Response**: First of all, the cavity flow and the KS traveling dynamics are quasi-periodic. Our setup assumes that the ODE solutions lie on a manifold $M$ embedded in $\mathbb{R}^n$, and by a manifold, we refer to specifically $d$-dimensional Riemannian manifolds. For general ODE, we do not foresee any advantage of this method since the vector fields $\mathbf{f}:\mathbb{R}^n\to \mathbb{R}^n$ do not admit any manifold structure although one can still numerically use the method with the proposed kernel induced by local tangent spaces of varying dimensions.

---

> > ### Comment · Reviewer_zhfF · 2024-11-25
> >
> > Thank you for the clarifications. I have no more questions.

---

> > > ### Author Response · Authors · 2024-11-26
> > >
> > > Thank you for your support and insightful comments that help us improved the manuscript.

---

### Official Review · Reviewer_7GGN · 2024-11-01

**Soundness:** 3
**Presentation:** 3
**Contribution:** 2
**Rating:** 5
**Confidence:** 5

**Summary:**

This paper definitely proposes an interesting method to learn the dynamics of a dynamical system that lives on a low-dimensional manifold based on point cloud-type data. The "geometrical constraint" refers to that the bases for building the kernel regression are projected to (an approximate) tangent space of the observations. The procedure to obtain the kernel regression's solution is modified to leverage of the (estimated) intrinsic dimension $d$. Then a "geometry-preserving" integrator is applied to get the trajectory based on the $\mathbf{f}_{\epsilon}$ learned in the kernel regression.

**Strengths:**

- Efficient compared with the baseline.

**Weaknesses:**

- Why just 1 baseline?
- Line 136: the writing on why the kernel regression opts for the L2 regularization and not the L1 regularization SINDy used is not very convincing.
- The paper only compares the end results with SINDy-based method. I suggest doing a more lateral comparison within the method itself.
  - Why not comparing the performance of different kernels such as Matern?
  - One of the most important aspect in this method is to compute the projection (matrix) onto the tangent space. Yet, this part is shoveled into the appendix. For example, the robustness is never mentioned, how the sensitive the approximation to the projection matrix affects the accuracy of the whole method? I say this because for multiple toy models in this paper, the representation of the manifold is known, then the analytic tangential projection matrix can be formed, which sets a perfect playground for such a study.
  - Since one has $\mathbf{f}_{\epsilon}$ once trained, why not comparing Euler+NC with a symplectic integrator (such as semi-implicit which shares similar spirit with this correction method)?


### Typos
- line 191: I am not sure if it is a typo but the squared exponential should not look like that, that is the Laplace kernel.

**Questions:**

N/A

---

> ### Author Response · Authors · 2024-11-25
> **Formal response - 1/2**
>
> Thank you for taking time to read the paper and your helpful comments.
>
> 1. Weakness, "Why just 1 baseline?"
>
> **Response**: In this revision, we provide two additional baseline methods: The Neural ODE (NODE) approach from Chen et al. 2018 and the Latent Dynamics Network (LDNet) from Regazzoni et al. 2024. See the updated Figure 1, where we reported the prediction for longer times (doubling the time in the previous report). The prediction for the entire time interval is shown in Fig 5. While the main discussion with added results can be found in Section 3.1 and Appendix E.1, let us highlight several new findings:
>
> + Euler+NC prediction is more accurate than NODE in all cases (see Table 4 for RMSE comparison). NODE fails to follow the manifold in a long-term prediction for the case $D=0.1$.
> + LDNet also fails to follow the manifold in a long-term prediction for $D=0.1$, but very accurate for $D=0.5$.
> + As we mentioned in the updated introduction (second paragraph), LDNet requires a latent variable of dimension larger than the intrinsic 1-dimension in this problem and sometimes produces solutions that do not lie on the manifold (as in the case of $D=0.1$).
> + For this example, we found that LDNet does not work if we use the same data set used for other methods (the reported results are obtained using twice the size of training data of the other methods).
>
> We also added these two baselines on the travelling waves example. Table 1 and Figure 4(j), (k) are updated, the paragraph discussing KS travelling wave in p.9 is also updated, and more detailed results and experiment configurations are presented in Appendix E.4 (see Fig 11). We found that both NODE and LDNet not only have high training costs, their predictions are less accurate compared to the proposed GMKRR.
>
> 2. Weakness, "Line 136 ..."
>
> **Response**: We agree with your concern. After a literature search, we found a paper Shi et al. 2019 suggesting to employ $\ell_q$ regularization for $0<q\leq 1$ in kernel regression. We adjusted this paragraph and cited the paper.
>
> 3. Weakness, "lateral comparison ..."
>
> 3(a). "different kernels such as Matern"
>
> **Response**: This is a good suggestion. In the revised manuscript, we added more results with the Matern kernel, as suggested (see Appendix E.1). We found that (at least for the simple example) there are no significant differences in terms of prediction skills compared to using the Gaussian kernel. This is as expected since this problem has smooth vector fields. There should be some examples of certain regularity where it is advantageous to employ the Matern kernel. However, we appreciate your suggestion, which confirms that the framework is not restrictive to only the Gaussian kernel.
>
> 3(b).  "... compute the projection (matrix) ..."
>
> **Response**: In the numerical examples, the only case where we know exactly the manifold is the motivating example in Section 3.1. While the cavity flow lies on a 2D torus, we have no idea what is the parameterization of this torus. For the remaining examples, the visualization of the manifolds obtained from the Isomaps are shown in panels (e)-(h), and just to remind you again, these coordinates are not used in our scheme to avoid dependence on other dimensionality reduction methods.
>
> Based on these facts, we provide a robustness study (as you suggested) on the motivating example in Section 3.1. See the new section ``Effect of Accuracy of Tangent Space'' in Appendix E.1.
> In this study, we verify the theoretical convergence rate for the normal correction step, that is the second component in the error bound in Theorem 3.1. We ignore checking the first error term, $\epsilon$, corresponds to the error in the estimation of the vector field, $\mathbf{f}$, since this error depends on cross validation and the regression method that is being used (in our case GMKRR).
>
> Assuming that the vector field is given, we found that the normal correction agrees reasonably with the theoretical convergence rates quite well (see Figure 7) and the discussion in p.26.

---

> ### Author Response · Authors · 2024-11-25
> **Formal response - 2/2**
>
> 3(c).  "... comparing Euler+NC with a symplectic integrator ..."
>
> **Response**: Based on the literature search, we believe that Leimkuhler et al 2000 is a reliable source for symplectic integrator on the Riemannian manifold. Upon reading this paper carefully, we found that, as the usual symplectic integrator, such method is designed to integrate Hamiltonian systems with a well-defined potential. Our system does not fit this configuration. The more important issue is that for general nontrivial Riemannian metrics, the symplectic scheme is semi-implicit, as you suggested. This requires one to solve a generating function (see Eq. (18)-(20) in Leimkuhler et al 2000) that involves not just the vector field but also the Riemannian metric and Christofel symbols, as shown in (17) in Leimkuhler et al 2000}. So, the method requires information more than just the estimated vector fields. While all these quantities can be approximated by our tools, the implicit nature of this scheme will require solving a nonlinear equation at each step, which is numerically not attractive.
>
> Since your suggestion is related to the second question from reviewers pB4e and u6B3, please read our response to their questions as well.
>
> 4. Typos:
>
> **Response**: Thanks for pointing out this typo. It is fixed.

---

> > ### Author Response · Authors · 2024-11-25
> > **References**
> >
> > + Ricky TQ Chen, Yulia Rubanova, Jesse Bettencourt, and David K Duvenaud. Neural ordinary differential equations. Advances in neural information processing systems, 31, 2018.
> > + B. Leimkuhler and G. W. Patrick. A symplectic integrator for riemannian manifolds. In Mechanics: From Theory to Computation, pages 239–256, New York, NY, 2000. Springer New York.
> > + Francesco Regazzoni, Stefano Pagani, Matteo Salvador, Luca Dede, and Alfio Quarteroni. Learning the intrinsic dynamics of spatio-temporal processes through latent dynamics networks. Nature Communications, 15(1):1834, 2024.
> > + Lei Shi, Xiaolin Huang, Yunlong Feng, and Johan AK Suykens. Sparse kernel regression with coefficient-based lq − regularization. Journal of Machine Learning Research, 20(161):1–44, 2019.

---

> > > ### Author Response · Authors · 2024-11-26
> > > **Follow up on the response**
> > >
> > > We hope this message finds you well. We are writing to follow up on our previous responses (and revised manuscript) and would like to check whether they address your questions.  If there are any remaining questions needing clarification, please feel free to let us know (especially since we are close to the deadline for revising the manuscript, which is tomorrow Nov 27th). We are more than happy to address them promptly.

---

> > > > ### Comment · Reviewer_7GGN · 2024-11-30
> > > >
> > > > Thanks for the response. Could you please specify more on how the bandwidths for different kernels are chosen? It reads a grid search is performed, yet the values chosen are not in the range later in the table.

---

> > > > > ### Author Response · Authors · 2024-11-30
> > > > > **About bandwidth selection**
> > > > >
> > > > > Thanks for the follow up question.  Indeed there is some missing information.  For the lateral comparison, after an initial grid search using the grid specified in the paper, we obtain an initial choice of hyperparameters $(\gamma^*, \ell^*, s^*)$. Surrounding this initial choice, we performed a random search to refine the hyperparameters using a computational budget of 100 trials.  The hyperparameters that produce the lowest RMSE are used for the prediction and listed in Table 5.
> > > > >
> > > > > See also the submitted codes in supplementary materials, `test/s1.py` for the grid search and `test/s1_lateral.py` for the random search procedure.
> > > > >
> > > > > If permitted by the editor, we can upload an updated manuscript with this detailed information added to Appendix E; otherwise, we can add this information in the final version (if the paper is accepted).
> > > > >
> > > > > Please let us know if you have any further questions.

---

> > > > > > ### Comment · Reviewer_7GGN · 2024-12-02
> > > > > >
> > > > > > If that is the case, the magnitude of $\gamma$ in the cases with a relative more complex dynamics indicates that it might be something else contributing other than the presented GMKRR. I decided to adhere to my original score as a data point. Thanks for the response.

---

> > > > > > > ### Author Response · Authors · 2024-12-03
> > > > > > >
> > > > > > > **Explanation of large bandwidth parameters**
> > > > > > >
> > > > > > > The larger values of the kernel bandwidth parameter for high-dimensional problems are effectively due to the scaling of the pair-wise L2 distances of the dataset (which can be overcome with the kernel $\tilde{\rho}$ in Appendix B.2).
> > > > > > >
> > > > > > > Here, perhaps your confusion is due to our abuse of notation on $\gamma$ in Appendix B.2. Specifically, Appendix B.2 reports the use of two similar kernels for different purposes.  The first kernel, "modified SE kernel", that we stated is $\tilde{\rho}(x,x';\gamma) = \exp(-\|x-x'\|^2 /(L^2\gamma))$, where L denotes the mean of pair-wise L2 distances of the dataset. This kernel is used for only the estimation of the intrinsic dimension.  To be more clear, we should have denoted the $\gamma$ in this kernel with another notation, such as $\eta$, and clarify that this first kernel that we are using is,
> > > > > > >
> > > > > > > $$\tilde{\rho}(x,x';\eta) = \exp(-\|x-x'\|^2 /(L^2\eta)).$$
> > > > > > >
> > > > > > > Subsequently, the second kernel in Appendix B.2 in the last paragraph refers to the one used in GMKRR (i.e., for learning the dynamics).  In other words, using the modified notation above, we empirically estimate
> > > > > > >
> > > > > > > $$\gamma=10(L^2\tilde{\eta})^{1/\tilde{d}}$$
> > > > > > >
> > > > > > > where this estimate is for the SE kernel in GMKRR as $\rho(x,x';\gamma)=\exp(-\|x-x'\|^2 /\gamma)$.  The actual values of the bandwidth parameters are reported in Table 6.  As a rule of thumb, for the Reaction-Diffusion problem $L^2 \approx 36931.0, \eta\approx 0.284$ and for the KS travelling waves, $L^2 \approx 6659.98, \eta \approx 0.210$.  This explains why the $\gamma$'s in Table 6 appear so large.
> > > > > > >
> > > > > > > **Some typos**
> > > > > > >
> > > > > > > We now have also identified a few more typos (abuse of notation) in Appendix B.2.  First, in the second equation, we missed a factor of 1/2, that is,
> > > > > > >
> > > > > > > $$d/2 \approx ... \equiv \max_\eta \tilde{d}(\eta)/2,$$
> > > > > > >
> > > > > > > so that $\tilde{d}$ is an intrinsic dimension estimated using $\tilde{\rho}(x,x';\eta)$ for a value of $\eta$.
> > > > > > >
> > > > > > > Second, we should clarify that we use the kernel $\tilde{\rho}$ and minimize over $\eta$ so we can clarify that we are minimizing,
> > > > > > >
> > > > > > > $$\frac{d}{2} \approx \max_\eta \frac{d\log(S(\eta))}{d\log\eta} = \max_\eta \frac{\tilde{d}(\eta)}{2},$$
> > > > > > >
> > > > > > > For more clear notations, refer to the maximum as $d^* = \tilde{d}(\eta^*)$ for the estimate of the dimension d. Here, $\eta^*$ denotes the argument of the optimization above. To employ the cross-validation, we empirically set a grid search around,
> > > > > > >
> > > > > > > $$\gamma^* = 10 (L^2 \eta^*)^{1/d^*}.$$
> > > > > > >
> > > > > > > For example, $0.8\gamma^*$, $1.0\gamma^*$, $1.2\gamma^*$, etc.  Yet, it was found that choosing $\gamma=\gamma^*$ for the SE kernel is sufficient for accurate GMKRR predictions.
> > > > > > >
> > > > > > > **Summary**
> > > > > > >
> > > > > > > We hope this convinces you that the method works not because of other things as you suspected. If there are more questions, please let us know as we found that all of these questions have been very helpful to clarify the presentation.  And we will make sure to update the manuscript in a future version.

---

### Official Review · Reviewer_u6B3 · 2024-11-04

**Soundness:** 3
**Presentation:** 3
**Contribution:** 2
**Rating:** 6
**Confidence:** 3

**Summary:**

Authors propose a new method for solving ODEs on manifolds in the multidimensional case which is based on the geometrically constrained operator-valued kernel and a geometry-preserving ODE solver. The efficiency and advantages of the method are demonstrated experimentally for a number of multidimensional problems and in comparison with alternative approaches.

**Strengths:**

1. The presentation of the material in the text of the manuscript is consistent and neat.

2. The problem of solving such a class of equations is relevant for a wide range of applications.

3. It has been empirically demonstrated that the proposed approach significantly outperforms known baselines (SINDy methods, CANDyMan) in both accuracy and speed on several classes of problems.

**Weaknesses:**

Overall, the work appears to be of high quality and relevant, however I would like to clarify the issues listed below in the "Questions" section.

**Questions:**

1. In the abstract you note that "machine learning methods ... struggle with ... lack of interpretability". It is not entirely clear from the text of the manuscript whether your method solves this particular problem.

2. Can you comment on whether NC Euler method (from sec. 3.2) is new and what alternative approaches exist for predicting dynamics on complex manifolds (if as you note RK2 method behaves badly here)?

3. Section "Related work" lacks a description of the essence of CANDyMan approach.

4. Is it possible to consider more baselines, e.g., method from the work "Neural Manifold Ordinary Differential Equations", described in "Related work"?

5. There are probably a few typos. Line 074: "We supplement the papers with..." (probably should be singular, i.e., "paper"); Line 446 and 475: "submanifold manifold" (probably word "manifold" is redundant here); Line 449: "the cost ... faster" (maybe cost is lower or training is faster).

---

> ### Author Response · Authors · 2024-11-25
> **Formal response**
>
> Thank you for taking time to read the paper and your helpful comments.
>
> 1. Questions, "In the abstract you note that ... interpretability ..."
>
> **Response**: We agree that the current method also does not provide interpretability; we remove this sentence from the abstract.
>
> 2. Questions, "Can you comment on whether NC Euler method ..."
>
> **Response**: This is a good point. In the revised manuscript (see the first paragraph in Section 3), we provide a more complete discussion of existing methods and motivate the proposed approach, putting it in the context of classical methods.
>
> 3. Questions, Section "Related work" lacks a description of the essence of CANDyMan approach.
>
> **Response**: We added a sentence in the third paragraph of Section 4 to emphasize CANDyMan's basic idea.
>
> 4. Questions, "Is it possible to consider more baselines ..."
>
> **Response**: As for the Neural Manifold ODE described in related work, Lou et al 2020, while their formulation can be used for general manifolds, the available code is tailored to specific manifolds (sphere, hyperboloid) with analytical exponential and log maps.  Furthermore, the code is developed for tasks such as density estimation, and a neural ODE is trained to match a target distribution at an end time starting from a given (simpler) distribution; this is entirely different from our problem, that aims to train the model to match a target trajectory over the entire time interval of interest, starting from a given initial condition.
>
> One possible idea from your suggestion is to adopt our manifold parameterization tools to extend their code, which is interesting, but this will require significant effort and warrant future investigation beyond the scope of this work. Second, while we believe this idea can potentially be fruitful, it is computationally costly as it involves training a nonlinear estimator with more parameters to be tuned compared to the proposed linear estimator.
>
> In this revision, we provide two additional baseline methods: The Neural ODE (NODE) approach from Chen et al. 2018 and the Latent Dynamics Network (LDNet) from Regazzoni et al. 2024. See the updated Figure 1, where we reported the prediction for longer times (doubling the time in the previous report). The prediction for the entire time interval is shown in Fig 5. While the main discussion with added results can be found in Section 3.1 and Appendix E.1, let us highlight several new findings:
>
> + Euler+NC prediction is more accurate than NODE in all cases (see Table 4 for RMSE comparison). NODE fails to follow the manifold in a long-term prediction for the case $D=0.1$.
> + LDNet also fails to follow the manifold in a long-term prediction for $D=0.1$, but very accurate for $D=0.5$.
> + As we mentioned in the updated introduction (second paragraph), LDNet requires a latent variable of dimension larger than the intrinsic 1-dimension in this problem and sometimes produces solutions that do not lie on the manifold (as in the case of $D=0.1$).
> + For this example, we found that LDNet does not work if we use the same data set used for other methods (the reported results are obtained using twice the size of training data of the other methods).
>
> We also added these two baselines on the travelling waves example. Table 1 and Figure 4(j), (k) are updated, the paragraph discussing KS travelling wave in p.9 is also updated, and more detailed results and experiment configurations are presented in Appendix E.4 (see Fig 11). We found that both NODE and LDNet not only have high training costs, their predictions are less accurate compared to the proposed GMKRR.
>
> 5. Question, typos
>
> **Response**: Thanks for pointing out these typos. They are fixed.

---

> > ### Author Response · Authors · 2024-11-25
> > **References**
> >
> > + Ricky TQ Chen, Yulia Rubanova, Jesse Bettencourt, and David K Duvenaud. Neural ordinary differential equations. Advances in neural information processing systems, 31, 2018.
> > + Aaron Lou, Derek Lim, Isay Katsman, Leo Huang, Qingxuan Jiang, Ser Nam Lim, and Christopher M De Sa. Neural manifold ordinary differential equations. Advances in Neural Information Processing Systems, 33:17548–17558, 2020.
> > + Francesco Regazzoni, Stefano Pagani, Matteo Salvador, Luca Dede, and Alfio Quarteroni. Learning the intrinsic dynamics of spatio-temporal processes through latent dynamics networks. Nature Communications, 15(1):1834, 2024.

---

> > > ### Author Response · Authors · 2024-11-26
> > > **Follow up on responses**
> > >
> > > We hope this message finds you well. We are writing to follow up on our previous responses (and revised manuscript) and would like to check whether they address your questions.  If there are any remaining questions needing clarification, please feel free to let us know (especially since we are close to the deadline for revising the manuscript, which is tomorrow Nov 27th). We are more than happy to address them promptly.

---

> > > > ### Comment · Reviewer_u6B3 · 2024-12-02
> > > >
> > > > Dear authors, I thank you for your response and revision of your work. Your explanations regarding the experimental part are clear. I consider it possible to raise my rating (5 -> 6) for the updated version.

---

> > > > > ### Author Response · Authors · 2024-12-02
> > > > >
> > > > > Thank you for your suggestions that have helped us improve the manuscript, your time to check the revision, and your support for the work with the new score.

---

### Official Review · Reviewer_pB4e · 2024-11-09

**Soundness:** 3
**Presentation:** 4
**Contribution:** 4
**Rating:** 10
**Confidence:** 4

**Summary:**

The authors propose a kernel method for learning and integrating ordinary differential equations on manifolds. The main contribution is the method for learning ODEs, called Geometrically constrained Multivariate Kernel Ridge Regression Method (GMKRR), a method that relies on the theory of Reproducing Kernel Hilbert Spaces. As for integrating, the author propose a new Euler method with a Normal direction correction.

**Strengths:**

The paper is very well written. The topic relies heavily on results from operator kernels, but I found that the introduction to the topic by the authors was adequate.
The contributions of the authors are novel and have promising applications in data-driven discovery of latent dynamics.
The experimental evaluation is good.

**Weaknesses:**

The method involves solving very large systems of equations at each step, and thus is computationally expensive. Nevertheless, the initial training time, (if existent, for instance when solving the system with a Cholesky decomposition) is much less compared with a PINN.
Furthermore, the timings are superior when compared with other data-driven methods, such as CANDyMan.

**Questions:**

Instead of the Euler-NC method, have you considered a second order Euler method for the ODE, by differentiating the kernel operator?

---

> ### Author Response · Authors · 2024-11-13
> **question**
>
> First, thank you for your support on the paper. We want to clarify your question before we respond to that. By second order Euler method, are you referring to the so-called Taylor' method? For example, to solve $$x'=f(x)$$, the method you are referring to is:
> $$
> X_{i+1} = X_i + h f(X_i) + \frac{h^2}{2} f'(X_i),
> $$
> and since our f is a kernel function, we need to take derivative of the kernel function with respect to time.

---

> ### Comment · Reviewer_pB4e · 2024-11-14
>
> Yes, that's what I mean, although the formula is not quite right. The correct formula is:
> $$X_{i+1}=X_{i} + h f(X_i)+\frac{h^2}{2} J_f(X_i) f(X_i),$$
> where $J_f$ stands for the Jacobian of $f$.
> The derivative of the kernel function would automatically learn the curvature of the manifold and would replace that last part estimated with GMLS.

---

> ### Author Response · Authors · 2024-11-25
> **Formal response**
>
> 1. The comment on weakness, "The method involves solving very large systems ..."
>
> **Response**: We clarify the computational cost by discussing the complexity of each prediction step in p.6 in a new Appendix C. Related to this issue, we also added a new Appendix A.4, discussing how to circumvent the computational bottleneck of realizing the normal vectors in (18) when the ambient dimension, $n$, is very high as shown in one of our numerical examples, $n=O(10^5)$.
>
> 2. Questions, "Instead of the Euler-NC method ..."
>
> **Response**: While it is possible to implement the second-order Euler method as you suggested, here are our thoughts.
> Practically, the implementation of the suggested method requires the time derivative of $\mathbf{f}_\epsilon$, and its computational cost
>
> $$\frac{d}{dt}\mathbf{f}\_\epsilon = (D\mathbf{f}\_\epsilon) \mathbf{f}_\epsilon,\quad (1)$$
>
> can be quite cumbersome since the estimator involves $\hat{\mathbf{T}}_{\mathbf{x}}$,
>
> $$\mathbf{f}\_\epsilon (\mathbf{x}) = \hat{\mathbf{T}}\_{\mathbf{x}} \mathbf{r}(\mathbf{x},X)\boldsymbol{\alpha},$$
>
> where $\mathbf{r}(\mathbf{x},X)$ also depends on $\hat{\mathbf{T}}\_{\mathbf{x}}$ and $X = \{\mathbf{x}\_1,\ldots, \mathbf{x}_N\}$.
>
> If $\mathbf{f}_\epsilon:\mathbb{R}^n\to\mathbb{R}^n$ is an analytic function, then one can approximate,
> Eq. (1) above with, e.g., a complex-step derivative approximation, such as,
>
> $$
> \frac{d}{dt}\mathbf{f}\_\epsilon(\mathbf{x})  =\text{Im}\left(\frac{\mathbf{f}\_\epsilon(\mathbf{x}+ i\delta\mathbf{f}_\epsilon(\mathbf{x}))}{\delta}\right) + O(\delta^2).
> $$
>
> where $i$ is the unit imaginary number. In our case $\mathbf{f}\_\epsilon: M \to \mathbb{R}^n$, one has to define the evaluation at $\mathbf{x}+ i\delta\mathbf{f}\_\epsilon(\mathbf{x})$ carefully since this vector is not in $M$, it is in $T\_{\mathbf{x}}M$. Second, even if this can be done, this approximation requires an additional cost from the evaluation of $\mathbf{f}\_\epsilon$ at the new point other than $\mathbf{x}$, which is documented above. On the other hand, with the proposed method, we readily have $\hat{\mathbf{T}}\_{\mathbf{x}}$ and $\hat{\mathbf{N}}\_{\mathbf{x}}$ whenever we need to evaluate $\mathbf{f}\_\epsilon$ at the base point $\mathbf{x}$, the extra computation is to fit GMLS to attain $\hat{g}$ and evaluate it on the vector field, $\hat{\mathbf{T}}\_{\mathbf{x}}^\top \mathbf{f}\_\epsilon$.
>
> Conceptually, the derivative of $\mathbf{f}_\epsilon$ will approximate the second fundamental form noted in (30) of the paper. On the other hand, $\hat{g}$ approximates $g$ in (32), which includes the second fundamental form and also the higher-order terms in the expansion. This suggests that the normal correction should in principle gives a more accurate result compared to the suggested improved Euler scheme.
>
> Please also see our response to Q2 of reviewer u6B3 and Q3(c) of reviewer 7GGN, which  are relevant to this issue.

---

> > ### Author Response · Authors · 2024-11-26
> > **follow up on our response**
> >
> > We hope this message finds you well. We are writing to follow up on our previous responses (and revised manuscript) and would like to check whether they address your questions.  If there are any remaining questions needing clarification, please feel free to let us know (especially since we are close to the deadline for revising the manuscript, which is tomorrow Nov 27th). We are more than happy to address them promptly.

---

### Meta-Review · Area_Chair_t73Z · 2024-12-17

**Metareview:**

The paper presents a novel method for learning vector fields of ordinary differential equations on manifolds by introducing a geometrically constrained operator-valued kernel and a geometry-preserving ODE solver. The proposed approach effectively addresses computational challenges in high-dimensional systems while ensuring solutions remain on or close to the manifold. The method is theoretically grounded, with demonstrated improvements in long-term prediction accuracy and efficiency across multiple examples such as cavity flow and reaction-diffusion dynamics. Reviewers provided constructive observations, noting the merit of the normal correction step in the solver, which improves results compared to explicit schemes. While some concerns were raised about underlying mechanics and kernel choices, the authors have adequately addressed these through clarifications and additional comparisons. The experimental results, robustness checks, and sound theoretical analysis collectively support the paper’s acceptance as a strong contribution, warranting a poster presentation at the conference.

**Additional Comments On Reviewer Discussion:**

During the reviewer discussion, key points included the robustness of the tangent space approximation, the comparison with additional baselines, and the kernel choice for the proposed method. 7GGN raised concerns about the sensitivity of the tangent projection, which the authors addressed with a robustness study verifying theoretical convergence rates. pB4e and u6B3 suggested expanding comparisons to methods like Neural ODE and LDNet; the authors incorporated these baselines, demonstrating the superiority of their approach in accuracy and efficiency. 7GGN also questioned kernel performance, prompting the inclusion of results with the Matern kernel, which showed no significant differences. Overall, the authors’ detailed responses and revisions effectively addressed the concerns, solidifying confidence in the paper’s contributions and leading to the final recommendation for acceptance.

---

### Decision · Program_Chairs · 2025-01-22

Accept (Spotlight)